# Microglia aging in the hippocampus advances through intermediate states that drive activation and cognitive decline

Jeremy M Shea[1]*, Saul A Villeda[1,2,3]*

[1]Department of Anatomy, University of California, San Francisco, San Francisco, United States; [2]Department of Physical Therapy and Rehabilitation Science, University of California San Francisco, San Francisco, United States; [3]Bakar Aging Research Institute, University of California, San Francisco, San Francisco, United States

### eLife Assessment

This **important** work advances our understanding of the aging trajectory and heterogeneity of hippocampal microglia. The authors provide an in-depth characterization of microglia in young and old mice as well as at intermediate time points, which reveals the existence of intermediate states characterized by a distinct transcriptional signature. The experimental approach is **solid**, especially with the validation of scRNA-seq findings with other methods. The study should be of interest to neuroimmunologists and biologists interested in aging.

*For correspondence:
jeremy.shea@ucsf.edu (JMS);
saul.villeda@ucsf.edu (SAV)

**Abstract** During aging, microglia – the resident macrophages of the brain – exhibit altered phenotypes and contribute to age-related neuroinflammation. While numerous hallmarks of age-related microglia have been elucidated, the progression from homeostasis to dysfunction during the aging process remains unresolved. To bridge this gap in knowledge, we undertook complementary cellular and molecular analyses of microglia in the mouse hippocampus across the adult lifespan and in the experimental aging model of heterochronic parabiosis. Single-cell RNA-Seq and pseudotime analysis revealed age-related transcriptional heterogeneity in hippocampal microglia and identified intermediate states of microglial aging that also emerge following heterochronic parabiosis. We tested the functionality of intermediate stress response states via TGFβ1 and translational states using pharmacological approaches in vitro to reveal their modulation of the progression to an activated state. Furthermore, we utilized single-cell RNA-Seq in conjunction with in vivo adult microglia-specific *Tgfb1* conditional genetic knockout mouse models to demonstrate that microglia advancement through intermediate aging states drives transcriptional inflammatory activation and hippocampal-dependent cognitive decline.

## Introduction

Microglia are one of few immune cells with long-term residence in the brain parenchyma (**Waisman et al., 2015**). Microglia support neural development by synaptic pruning and neurogenic function in the developing brain, followed by homeostatic maintenance in the adult brain (**Salter and Beggs, 2014**). Alternatively, in the aged brain microglia exhibit unbalanced phenotypes, aggravating perceived insults rather than responding productively to environmental signals (**Mosher and Wyss-Coray, 2014**). While microglia have been identified as the principal mediators of multiple age-related

neurodegenerative pathologies (*Bachiller et al., 2018*), the mechanisms leading to microglia dysfunction during aging remain obscure.

Numerous studies have indicated that aged microglia are inflamed (*Sierra et al., 2007*), have reduced phagocytic capacity (*Marschallinger et al., 2020*; *Pluvinage et al., 2019*), and have decreased motility (*Damani et al., 2011*). Microglia exhibit several hallmarks of aging that potentially contribute to their age-related dysfunction, such as shortened telomeres (*Flanary et al., 2007*), altered intercellular communication (*Udeochu et al., 2016*), molecular alterations (*Grabert et al., 2016*), and a loss of proteostasis (*Burns et al., 2020*). Furthermore, many recent studies have started to reveal the molecular changes that define microglial aging. Single-cell RNA-Seq (scRNA-Seq) analyses indicate that microglia isolated from the entire brain lose homeostasis and activate inflammatory transcriptional profiles with age (*Hammond et al., 2019*; *Li et al., 2023*). Data-rich studies have also revealed partial overlaps between aging microglia and those from disease models, including Alzheimer's disease (*Silvin et al., 2022*; *Keren-Shaul et al., 2017*; *Sala Frigerio et al., 2019*; *Lee et al., 2023*; *Masuda et al., 2019*). Interestingly, the microglial response to aging shows regional variation as white matter-rich regions induce an interferon signature in aged microglia (*Kaya et al., 2022*). Studies using aged plasma administration and heterochronic blood exchange demonstrate that microglia aging is in part driven by the aged systemic environment (*Yousef et al., 2019*; *Rebo et al., 2016*). However, the genesis of age-related microglial phenotypes has not been extensively investigated. So, we set out to characterize the progression of age-related hippocampal microglial changes, aiming to uncover intermediate states that could be intrinsic to the aging process. To do so, we undertook complementary cellular and molecular analyses of microglia across the adult lifespan and in heterochronic parabiosis – an experimental model of aging in which the circulatory systems of young adult and aged animals are joined (*Villeda et al., 2011*).

In this study, we report that microglia in the adult mouse hippocampus, a brain region responsible for learning and memory and susceptible to age-related cognitive decline, advance through intermediate states that drive inflammatory activation during aging. We utilize scRNA-Seq across the adult lifespan to identify intermediate transcriptional states of microglial aging that emerge following exposure to an aged systemic environment. We observe heterogeneous spatiotemporal dynamics of microglia activation across the adult lifespan with pseudotime and immunohistochemical analysis. Functionally, we tested the role of these intermediate states using in vitro microglia approaches and in vivo temporally controlled adult microglia-specific *Tgfb1* conditional genetic knockout mouse models to demonstrate that intermediates represent modulators of the progression of microglia from homeostasis to activation, with functional implications for hippocampal-dependent cognitive decline.

# Materials and methods

**Key resources table**

| Reagent type (species) or resource | Designation | Source or reference | Identifiers | Additional information |
|---|---|---|---|---|
| Strain, strain background (*Mus musculus*) | C57BL/6J | Jackson Laboratories | RRID:IMSR_JAX:000664 | |
| Strain, strain background (*M. musculus*) | Cx3cr1^Cre-ER | Jackson Laboratories | RRID:IMSR_JAX:021160 | |
| Strain, strain background (*M. musculus*) | Tmem119^Cre-ER | Jackson Laboratories | RRID:IMSR_JAX:031820 | |
| Strain, strain background (*M. musculus*) | Tgfb1-flox | Jackson Laboratories | RRID:MMRRC_065809-JAX | |
| Commercial assay or kit | Neural Dissociation Kits (P) | Miltenyi | 130-092-628 | |
| Commercial assay or kit | Myelin Removal Beads | Miltenyi | 1130-096-731 | |
| Antibody | CD11b-APC | eBioscience | 17-0112-82, RRID:AB_469343 | FACS 1:200 |
| Antibody | CD45-PE | eBioscience | 12-0451-82, RRID:AB_465668 | FACS 1:200 |

*Continued on next page*

Continued

| Reagent type (species) or resource | Designation | Source or reference | Identifiers | Additional information |
|---|---|---|---|---|
| Antibody | CD48-Pacific Blue | BioLegend | 103417, RRID:AB_756139 | FACS 1:200 |
| Software, algorithm | DeSeq2 (1.42.1) | https://bioconductor.org/packages/release/bioc/html/DESeq2.html | RRID:SCR_015687 | |
| Commercial assay or kit | Cd11b magnetic beads | Miltenyi | 130-126-725 | |
| Software, algorithm | Cell Ranger (v5.0.1) | 10x Genomics | RRID:SCR_021160 | |
| Software, algorithm | Seurat (v3.2.1) | https://satijalab.org/seurat/articles/install_v5 | RRID:SCR_007322 | |
| Software, algorithm | Monocle3 (v1.3.4) | https://cole-trapnell-lab.github.io/monocle3/docs/installation/ | RRID:SCR_018685 | |
| Software, algorithm | Scorpius (v1.0.9) | https://github.com/rcannood/SCORPIUS; *Cannoodt, 2021* | | |
| Antibody | Anti-Iba1, rabbit | Wako | 019-19741, RRID:AB_839504 | 1:1000 IHC |
| Antibody | Anti-CD68, rat | Bio-Rad | MCA1957, RRID:AB_322219 | 1:250 IHC |
| Antibody | Anti-NFKB p65, rabbit | SantaCruz | sc-372, RRID:AB_632037 | 1:500 IHC |
| Antibody | Anti-Iba1, guinea pig | Synaptic Systems | 234-004, RRID:AB_2493179 | 1:1000 IHC |
| Antibody | Anti-C1q, rabbit | Abcam | ab182451, RRID:AB_2732849 | 1:1000 IHC |
| Antibody | Anti-C3, rat | Abcam | ab11862, RRID:AB_2066623 | 1:1000 IHC |
| Antibody | Anti-S6, rabbit | Cell Signaling | 2217, RRID:AB_331355 | 1:500 IHC |
| Antibody | Anti-KLF2, rabbit | Bioss | bs-2772R, RRID:AB_10857057 | 1:250 IHC |
| Peptide, recombinant protein | M-CSF | Peprotech | 315-02 | 10 ng/mL |
| Peptide, recombinant protein | TGFβ1 | Thermo Fisher Scientific | PHG9204 | 10 ng/mL |
| Chemical compound, drug | CX-5461 | EMD Millipore | 509265 | 100 nM |
| Sequence-based reagent | Tgfb1 RNAScope Probe | ACD | 443571C2 | |
| Software, algorithm | ggplot2 | https://ggplot2.tidyverse.org/ | RRID:SCR_014601 | |
| Software, algorithm | Prism 8.0 | GraphPad | RRID:SCR_002798 | |

## Mice

All animal procedures were performed in accordance with the protocols approved by the UCSF IACUC. Animals were housed in SPF barrier facilities and provided continuous food and water along with environmental enrichment. Aging characterizations were performed on an inhouse C57BL/6J mouse colony where 2-month-old mice were purchased from Jackson Laboratories (Stock 000664) and aged in the UCSF Parnassus barrier facility. Timed pregnant females were obtained for isolation of primary microglia from newborn pups. Male mice were used in all experiments. B6.129P2(Cg)-*Cx-3cr1*tm2.1(cre/ERT2)Litt/WganJ (*Cx3cr1*Cre-ER) (Stock 021160), C57BL/6-*Tmem119*em1(cre/ERT2)Gfng/J (*Tmem-119*Cre-ER) (Stock 031820), and C57BL/6J-Tgfb1em2Lutzy/Mmjax (*Tgfb1*flox) (Stock 65809-JAX) mice were obtained from Jackson Laboratories. Mice were bred to obtain *Cx3cr1*Cre-ER/+ or *Tmem119*Cre-ER/+ with either *Tgfb1*wt/wt, *Tgfb1*fl/wt, or *Tgfb1*fl/fl. As *Cx3cr1*Cre-ER is a knock-in/knock-out allele for the microglia homeostatic gene *Cx3cr1*, all mice tested were *Cx3cr1*Cre-ER +/wt. To induce recombination and deletion of *Tgfb1* by CRE-ER, we injected tamoxifen for 5 days at 90 mg/kg daily, and mice were analyzed after 60 days.

## Parabiosis

Parabiosis surgery followed previously described procedures (*Smith et al., 2015*). Mirror-image skin incisions at the left and right flanks were made through the skin, and shorter incisions were made through the peritoneum. The peritoneal openings of the adjacent parabionts were sutured together with chromic gut suture (MYCO Medical, GC635-BRC). Apposing elbow and knee joints from each parabiont were sutured together (Coated VICRYL Suture, Ethicon, J386) and the skin of each mouse was stapled (9 mm Autoclip, Clay Adams, 427631) to the skin of the adjacent parabiont. Each mouse was injected subcutaneously with Carpofen, Enrofloxacin, and Buprenex as directed for pain and monitored during recovery. For overall health and maintenance behavior, several recovery characteristics were analyzed at various times after surgery, including pair weights and grooming behavior.

## Bulk microglia RNA-Seq

Microglia were isolated from C57BL6/J, *Tgfb1* cHet, and *Tgfb1* cKO using FACS. Briefly, mice were sedated with ketamine followed by perfusion with 30 mL of ice-cold phosphate buffered saline (PBS). The entire brain was removed, then the hippocampus was sub-dissected. Single-cell suspensions were generated by enzyme-mediated (papain) and mechanical dissociation using Miltenyi Neural Dissociation Kits (P) (Miltenyi, 130-092-628) according to the manufacturer's instructions. Myelin was depleted from the suspensions using Myelin Removal Beads (Miltenyi, 130-096-731). Cells were labeled with CD11b-APC (eBioscience, 17-0112-82, RRID:AB_469343) and CD45-PE (eBioscience, 12-0451-82, RRID:AB_465668), then sorted at 4°C into Tri Reagent with a FACS AriaII (BD). The gating strategy for microglia isolation is to collect Cd11b$^+$Cd45$^{Intermediate}$ cells. During the collection of control and *Tgfb1* microglia, CD48 was detected in Cd11b$^+$Cd45$^{Intermediate}$ cells using CD48-Pacific Blue (BioLegend, 103417, RRID:AB_756139). FlowJo was used to analyze flow cytometry data. RNA was isolated from microglia using Tri Reagent (Sigma-Aldrich, T9424). RNA pellets were resuspended in 5 uL TE buffer.

RNA was transformed into RNA-Seq libraries using an adapted version of the Smart-Seq2 protocol (*Trombetta et al., 2014*). Briefly, 5 ng of total RNA was reverse transcribed with SuperScript II (Thermo Fisher, 18064014) supplemented with betaine and MgCl$_2$ using an oligo-dT RT primer (AAGC AGTGGTATCAACGCAGAGTACT(30)VN) with a PCR binding site and a template switching oligonucleotide (5′ AAGCAGTGGTATCAACGCAGAGTACATrGrG+G) with a homotypic PCR binding site. Betaine and MgCl2 were added to enhance the reaction. After reverse transcription, whole transcriptome amplification by PCR was performed for 12 cycles using KAPA HiFi HotStart ReadyMix (Roche, 7958935001) with a PCR primer (AAGCAGTGGTATCAACGCAGAGT). The PCR reaction was cleaned up with Ampure XP beads (Beckman Coulter, A63881). In addition, after the whole transcriptome PCR amplification step, qPCR was performed to determine the presence of microglia-specific transcripts (*Cx3cr1*) and housekeeping genes (*Gapdh*). The amplified DNA was diluted to a concentration of 0.5 ng/uL and subjected to tagmentation with the Illumina Nextera XT kit (Illumina, FC-131-1096). Each sample was PCR amplified with a unique set of Nextera indices. Bulk microglia RNA-Seq libraries were sequenced on an Illumina HiSeq, while in vitro microglia RNA-Seq libraries were sequenced on an Illumina Nova-Seq.

## RNA-Seq analysis

FASTQ reads were pseudo aligned to the mouse transcriptome (GRCm39 cDNA from Ensembl) using kallisto (*Bray et al., 2016*) with default parameters. Next, transcript abundance estimates were imported into DeSeq2 (*Love et al., 2014*). Differential expression analysis was performed using DeSeq2 using the Wald significance test. Gene ontology analysis was performed with Panther (*Ashburner et al., 2000*; *Carbon et al., 2021*; *Mi et al., 2019*). Principal component analysis (PCA) plots were generated with the plotPCA function included with DeSeq2 and graphed with ggplot2. Heatmaps were generated with the pheatmap package in R.

The overlap between gene expression differences in aging and lipopolysaccharides LPS treatment was determined using genes significantly changed between the 6- and 24-month-old timepoints in the scRNA-Seq data and the control and LPS-treated primary microglia. $\chi^2$ test was used to determine the significance of the overlap between these two datasets. The Venn diagram was generated using the VennDiagram package in R.

## 10x Genomics single-cell RNA-sequencing

10x single-cell RNA-Seq libraries were generated from Cd11b+ cells isolated from the hippocampi of an aging cohort of mice consisting of 6-, 12-, 18-, and 24-month-old C57BL/6J mice that were all collected and processed on the same day in an interspersed order. Mice were sedated with ketamine followed by perfusion with ice-cold PBS. Subsequently, the entire brain was removed from the mouse followed by sub-dissection of the hippocampus. For each age, hippocampi from five mice were pooled during the dissection step in HBSS at 4°C. Single-cell suspensions were generated by enzyme-mediated (papain) and mechanical dissociation using Miltenyi Neural Dissociation Kits (P). The papain dissociation was done at 37°C for 10 minutes with three trituration steps. All other processing steps were performed at 4°C. Myelin was depleted from the suspensions using Myelin Removal Beads (Miltenyi). Microglia were enriched using Cd11b magnetic beads (Miltenyi, 130-126-725). Cell viability was determined to be over 95%. Single-cell RNA-Seq 3' libraries were generated from the cell suspension using 10x Genomics Chromium Single-Cell 3' Solution. Libraries were sequenced on the Illumina Nova-Seq. Cell Ranger demultiplexed and mapped (using bcl2fastq) reads, followed by alignment (with STAR), and generation of single-cell expression matrices (*Zheng et al., 2017*).

The isolation protocol for the *Tgfb1* genetic mouse model was modified. We added transcriptional and translational inhibitors (Actinomycin D [Sigma-Aldrich, A1410], Anisomycin [Sigma-Aldrich, A9789], and Triptolide [Sigma-Aldrich, T3652]) to prevent transcriptional changes due to ex vivo activation (*Marsh et al., 2022*). We utilized the concentrations presented in *Marsh et al., 2022*.

## Single-cell RNA-Seq analysis
### Seurat
The single-cell gene expression matrices generated by Cell Ranger were loaded into Seurat (*Butler et al., 2018*).

## Aging microglia dataset

Cells were filtered according to the following parameters: genes >1000 & RNA count >10,000 & mitochondrial percentage <5. The count data was log normalized. The 2500 most variable genes were identified using the 'vst' method. The expression data was scaled and centered. PCA was performed, the first 25 PCAs were used to identify nearest neighbors, and interconnected clusters were identified with a resolution of 0.4. UMAP was used to visualize the results. Preliminary dimensionality reduction and visualization with UMAP using 25 PCAs and 25 nearest neighbors in Seurat identified a large microglia cluster, plus sparse clusters consisting of astrocytes, vascular cells, neutrophils, and macrophages. *Hexb*, *P2ry12*, *Tmem119*, and *Slc2a5* are used as markers to identify microglia in the preliminary clustering, while *Nrxn1* (neurons), *Tm4sf1* (vasculature), *Dclk*1 (astrocytes), *Sdpr* (vasculature smooth muscle), *Abcb1a* (pericyte/vasculature), and *Foxq1* (macrophages) identified other cell types. The data was then subsetted on the non-proliferating microglial clusters (to avoid confounds associated with the cell cycle). The top 1500 most variable genes in the subsetted microglia were identified using the 'vst' method. The data was scaled and centered. PCA was performed, the first eight PCAs were used to identify nearest neighbors, and interconnected clusters were identified with a resolution of 0.25. UMAP was used to visualize the results using 8 PCAs and 100 nearest neighbors. Gene enrichment and differential expression analysis were performed on the filtered dataset to determine the differences between ages.

## *Tgfb1* cKO microglia dataset

Cells were prefiltered according to the following parameters: Hexb > 4 & Tmem119 > 1 & Cx3cr1 > 4 & Top2a < 1 & Mki67 < 1 nFeature_RNA > 1000 & nFeature_RNA < 6000 & nCount_RNA > 2000 & nCount_RNA < 40,000 & percent.mt < 10. Data was integrated using IntegrateData and scaled. PCA was performed, the first 50 PCAs were used to identify nearest neighbors, and interconnected clusters were identified with a resolution of 0.35. UMAP was used to visualize the results. Preliminary dimensionality reduction and visualization with UMAP using 50 PCAs identified a large microglia cluster, plus clusters of other cell types. The non-proliferative microglia clusters were subsetted for further analysis (the macrophage cluster was differentiated from the microglia cluster based on increased expression of *Mcp1*, *Pf4*, and *Ms4a7*). The data was scaled and centered. PCA was performed, the first 10 PCAs

were used for UMAP dimensionality reduction and identify nearest neighbors, and interconnected clusters were identified with a resolution of 0.20.

## Monocle 3

Monocle 3 was used to generate the pseudotime analysis (*Qiu et al., 2017a*; *Qiu et al., 2017b*; *Trapnell et al., 2014*; *Cao et al., 2019*). Data was imported from Seurat analysis and a Monocle dataset was generated. The dataset was preprocessed with normalization of the data and PCA generation. Next, the preprocessed data underwent further non-linear dimensionality reduction using UMAP. Cells were clustered, and trajectories were determined for cells within a cluster. For determining 'pseudotime' aging of microglia, the root node of the pseudotime trajectory was manually placed in the cluster of 6-month cells. The graph segments of the pseudotime trajectories resulting in inflammatory activation were chosen for further analysis. Graph autocorrelation analysis was performed on the inflammatory trajectory using Moran's I, which determines if genes are expressed in focal regions of graph space. A cutoff q-value of 0.005 was set as significant for focal expression. Subsequently, coregulated modules were determined using Louvain community analysis at an optimized resolution of 0.0078. The modules were then overlaid onto the UMAP plot to determine the region of the trajectory with focal expression of the module. The 10 most significant genes of each module as determined by q-value were used to construct the heatmaps in Figure 3F and I. The significant genes of each module were used to determine the effects of manipulations in the presence of LPS in Figure 3J.

## Scorpius

Scorpius was used to generate a secondary pseudotime analysis that confirmed the results of the Monocle 3 analysis (*Saelens et al., 2019*). Data was imported from Seurat analysis as a SingleCellExperiment and dimensionality was reduced. A trajectory plot was generated followed by a trajectory inference. Candidate marker genes were identified using the gene_importances command. Subsequently, gene expression modules were generated with the extract_modules command and visualized using a heatmap.

## Immunohistochemistry (IHC)

Mice were sedated with ketamine followed by perfusion with 30 mL of ice-cold PBS. Brains were collected from PBS perfused mice and fixed overnight at 4°C in 4% paraformaldehyde in PBS. The brains were washed with PBS and immersed in 30% sucrose (in PBS), and then stored at 4°C until the brains sank. 40 µM coronal sections were sliced using a microtome at –20°C. Sections were stored in cryopreservation buffer (40% PB buffer [0.02 M sodium phosphate monobasic, 0.08 M sodium phosphate dibasic, pH 7.4], 30% glycerol, 30% ethylene glycol) at –20°C until staining. Hippocampal sections were washed three times in TBST. Sections were permeabilized in pretreatment buffer (0.1% Triton-X in TBST) with rocking for 1 hour at room temperature. After three washes in TBST, sections were blocked in 5% goat serum for 2 hours (for IBA1/CD68 double stain) or 5% donkey serum (NFKB/Iba1 and C1q/C3/Iba1 stains). Blocking buffer was replaced with primary antibody mixture and incubated overnight at 4°C with rocking. The following primary antibodies were used: rabbit anti-IBA1 (1:1000, Wako, Cat# 019-19741, RRID:AB_839504), rat anti-CD68 (1:250, Bio-Rad, Cat# MCA1957, RRID:AB_322219), rabbit anti-NFKB p65 (1:500, Santa Cruz, Cat# sc-372, RRID:AB_632037), guinea pig anti-Iba1 (1:1000, Synaptic Systems, Cat# 234-004, RRID:AB_2493179), rabbit anti-C1q (1:1000, Abcam, Cat# ab182451, RRID:AB_2732849), rat anti-C3 (1:1000, Abcam, Cat# ab11862, RRID:AB_2066623), rabbit anti-S6 (1:500, Cell Signaling, Cat# 2217, RRID:AB_331355), and rabbit anti-KLF2 (1:250, Bioss, Cat# bs-2772R, RRID:AB_10857057). The sections were washed three times with TBST. Subsequently, secondary antibody solution was added to the sections and incubated at room temperature with rocking for 2 hours. The following secondary antibodies were used: goat anti-rabbit AlexaFluor 488 (Life Technologies, Cat# A-11008, RRID:AB_143165), goat anti-rat AlexaFluor 555 (Life Technologies, Cat# A-21434, RRID:AB_2535855), donkey anti-guinea pig AlexaFluor 488 (Jackson Immuno, Cat# 706-545-148, RRID:AB_2340472), donkey anti-rabbit AlexaFluor 555 (Life Technologies, Cat# A-31572, RRID:AB_162543), and donkey anti-rat AlexaFluor 647 (Jackson Immuno, Cat# 712-606-150, RRID:AB_2340695). The sections were washed three times with TBST, with the first wash containing Hoescht 33342 (Invitrogen, H3570) at a concentration of 1 µg/mL. The sections were transferred to phosphate buffer and mounted on frosted slides. Coverslips were mounted with

Prolong Gold Antifade (Thermo Fisher, P10144) reagent after the sections were dried. Slides were imaged at ×20 magnification on a Zeiss LSM 800 or LSM900 for a final resolution of 0.312 μM/pixel. 8–9 z-planes were imaged at 3 μM intervals. The hippocampus was imaged, and the tiled images were stitched together with Zeiss ZenPro software. 2–3 hippocampal sections were imaged from each sample.

Images were analyzed with FIJI. Maximum intensity projections of Z-stacked images were constructed. Images were converted to 8-bit images and consistent adjustments were made for each channel within an experiment. The Iba1 channel was used to quantify microglia using the particle counter function in FIJI, and the counts were validated by manual visual counting. The presence of CD68, NFKB(p65), C3, or C1q in microglia was determined by creating regions of interest (ROIs) within the microglia in a field using the create selection function on the Iba1 images. The ROIs of the hippocampal subregions and the microglia were combined, and CD68 puncta meeting a minimum particle size threshold (100 pixels, equating to 31.2 μM) within microglia determined if the microglia were activated (CD68+IBA1+), while the measure function determined the intensity of NFKB(p65), C3, or C1q. To determine percent activation of microglia with IBA1/CD68 staining, we divided the number of microglia that were activated (CD68+IBA1+) by the total number of microglia (IBA1+) in the analyzed regions and multiplied by 100.

Additionally, we tested whether autofluorescence from aged tissue was interfering with our signal using TrueBlack Lipofuscin Autofluorescence Quencher (Cell Signaling, Cat# 92401). Using the IBA1/CD68 doublestain, we obtained very similar results for microglia activation. Furthermore, our KLF2 stain was performed with TrueBlack.

## In vitro treatment of microglia

Mixed glial cultures were generated from p1 C57BL/6J male pups. Cortices and hippocampi were isolated from the pups, and meninges were removed. The tissue was broken up by repeatedly pipetting up and down, followed by dissociation with Trypsin. Single-cell suspensions were plated on poly-L-lysine-coated flasks in Dulbeco's minimal eagle media (DMEM) (Thermo Fisher, 11965-118) supplemented with 10% fetal bovine serum (FBS) (Gemini Bio-Products, 900-208) and antibiotic mix (Penicillin/Streptomycin) (Genesee Scientific, 25-512) and incubated at 37°C in 5% $CO_2$. Media were changed after 4 days to remove debris. Microglia were isolated after 14 days by agitating the plates to dislodge microglia, then transferring the supernatant of each sample evenly among 4 wells of a 24-well plate. Microglia were incubated for 16 hours followed by replacement with serum-free macrophage SFM media (Thermo Fisher, 12065074) supplemented with M-CSF (10 ng/mL) (Peprotech, 315-02) and antibiotic mix. After 24 hours, microglia were treated with RNA polymerase I Inhibitor 2, CX-5461 (100 nM) (EMD Millipore, 509265) or human TGFβ1 recombinant protein (10 ng/mL) (Thermo Fisher Scientific, PHG9204) for 24 hours followed by the addition of LPS (200 ng/uL) from *Escherichia coli* (Sigma-Aldrich, L2018) for 8 hours and ATP (10 nM) (InvivoGen, tlrl-atpl) for 30 minutes. Tri Reagent was added to the culture plate, and the plates were shaken for 5 minutes. RNA was isolated according to the manufacturer's instructions. RNA-Seq libraries were generated as above for the bulk RNA-Seq.

## RNAscope

RNAscope was performed with the RNAscope Multiplex Fluorescent Reagent Kit v2 (ACD, 323110). 40 μM coronal sections were washed with tris buffered saline with Tween 20 (TBST), then incubated with hydrogen peroxide for 45 minutes at 25°C. After washing with TBST, the sections were incubated in Target Retrieval Buffer at 95°C for 10 minutes. The sections were mounted on slides and dried overnight. The sections were treated with RNAscope Protease III for 10 minutes at 40°C. Sections were washed in water four times. Sections were incubated with *Tgfb1* probe (ACD, 443571-C2) for 2 hours at 40°C. The sections were washed twice with RNAscope wash buffer. The sections were incubated with AMP1 for 30 minutes, AMP2 for 30 minutes, AMP3 for 15 minutes, HRP-C2 for 15 minutes, TSA Plus Cy3 (1:1000, PerkinElmer, SKU NEL744001KT) for 30 minutes, and HRP-Blocker for 15 minutes; all at 40°C with washes with RNAscope wash buffer between every incubation. RNA-Protein Co-Detection Ancillary Kit (ACD, 323180) buffer was used to block the sections before incubating with rabbit anti-IBA1 (1:500, Wako, Cat# 019-19741, RRID:AB_839504) overnight at 4°C. Sections were washed three times in TBST. Sections were incubated with secondary antibody solution with goat anti-rabbit AlexaFluor 488 (1:1000, Life Technologies, Cat# A-11008,

RRID:AB_143165) for 2 hours at room temperature. The sections were washed three times with TBST, with the first wash containing Hoescht 33342 (Invitrogen, H3570) at a concentration of 1 µg/mL. Coverslips were mounted with Prolong Gold Antifade (Thermo Fisher, P10144) reagent after the sections were dried. Slides were imaged at ×40 magnification on a Zeiss LSM 900. 10 z-planes were imaged at 1 µM intervals.

Images were analyzed with FIJI. Maximum intensity projections of Z-stacked images were constructed. Images were converted to 8-bit images and the *Tgfb1* was thresholded. For each section, the intensity of 5–7 individual *Tgfb1* puncta was averaged to get the average puncta intensity of that section. The intensity of all *Tgfb1* signal in clearly identifiable individual IBA1 cells was measured for 20–25 cells in the molecular layer of the hippocampus. The mean *Tgfb1* count per microglia for each sample was calculated by dividing the average *Tgfb1* intensity in IBA1 cells by the average puncta intensity.

## Contextual fear conditioning

In this task, mice learned to associate the environmental context (fear conditioning chamber) with an aversive stimulus (mild foot shock; unconditioned stimulus [US]) during the training phase enabling testing for hippocampal-dependent contextual fear conditioning. To also assess amygdala-dependent cued fear conditioning, the mild foot shock was paired with a light and tone cue (conditioned stimulus [CS]) during the training phase. Conditioned fear was displayed as freezing behavior. Specific training parameters are as follows: tone duration is 30 seconds; level is 70 dB, 2 kHz; shock duration is 2 seconds; intensity is 0.6 mA. This intensity is not painful and can easily be tolerated but will generate an unpleasant feeling. More specifically, on day 1 each mouse was placed in a fear-conditioning chamber and allowed to explore for 2 minutes before delivery of a 30-second tone (70 dB) and light ending with a 2-second foot shock (0.6 mA). 2 minutes later, a second CS-US pair was delivered. On day 2, each mouse was first placed in the fear-conditioning chamber containing the same exact context, but with no CS or foot shock. Freezing was analyzed for 2 minutes and represented as the percentage of time that the mouse froze over the 2 minutes. 1 hour later, the mice were placed in a new context containing a different odor, floor texture, chamber walls, and shape. Animals were allowed to explore for 2 minutes before being re-exposed to the CS. Freezing was analyzed for 30 seconds following the CS and represented as the percentage of time that the mouse froze over those 30 seconds. Determination of freezing behavior was performed using FreezeScan video tracking system and software (Cleversys, Inc). Single outliers were removed using the extreme studentized deviate method (Grubbs' test) with an alpha of 0.05.

## Novel object recognition

The novel object recognition task was adapted from a previously described protocol (*Leger et al., 2013*). Specifically, during the habituation phase (day 1), mice could freely explore an empty open-field arena for 10 minutes (motor activity and anxiety [time in center] were measured during this phase). During the training phase (day 2), two identical objects were placed in the habituated arena, and mice could explore the objects for 5 minutes. For the testing phase (day 3), one object was replaced with a novel object, and mice could explore the objects for 5 minutes. Time spent exploring each object was quantified using the Smart Video Tracking Software (Panlab; Harvard Apparatus). Two different sets of objects are used. To control for any inherent object preference, half of the mice are exposed to object A as their novel object and half to object B. To control for any potential object-independent location preference, the location of the novel object relative to the trained object is also varied. The objects were chosen based on their ability to capture the animal's interest, independent of genetic background or age. To determine percent time with novel object, we calculate (Time with novel object)/(Time with Trained Object + Time with Novel Object) * 100. In this preference index, 100% indicates full preference for the novel object, and 0% indicates full preference for the trained object. A mouse with a value of 50% would have spent equal time exploring both objects. Mice that did not explore both objects for 5 seconds during the training phase or testing phase were excluded from analysis. Single outliers were removed using the extreme studentized deviate method (Grubbs' test) with an alpha of 0.05.

## Y maze

The Y maze task was conducted using an established forced alternation protocol (*Belarbi et al., 2011*). During the training phase, mice were placed in the start arm facing the wall and allowed to explore the start and trained arm for 5 minutes, while entry to the third arm (novel arm) was blocked. The maze was cleaned between each mouse to remove odor cues, and the trained arm was alternated between mice. The mouse was then removed to its home cage. After 30 minutes, the block was removed, and the mouse was returned to the start arm and allowed to explore all three arms for 5 minutes. Time spent in each arm was quantified using the Smart Video Tracking Software (Panlab; Harvard Apparatus). The percent time in the novel arm was defined as time in the novel arm divided by time spent in the novel and trained arms during the task.

## Datasets

All RNA-Seq and scRNA-Seq data have been deposited in the Gene Expression Omnibus and are publicly available as of the date of publication. The following datasets were generated for this article and deposited in Gene Expression Omnibus: aging single-cell RNA-Seq (GSE179358), in vitro-treated primary microglia (GSE179611), *Tgfb1* cKO microglia RNA-Seq (GSE190007), and *Tgfb1* cKO microglia single-cell RNA-Seq (GSE211340). The following publicly available datasets were analyzed in this article: mouse model of Alzheimer's disease microglia single-cell RNA-Seq (GSE127892) (*Sala Frigerio et al., 2019*) and heterochronic parabiosis microglia single-cell RNA-Seq (GSE193093) (*Pálovics et al., 2022*).

## Statistics

Sample sizes were determined from performing power analysis based off previous results in the laboratory. A minimum sample size of n=3 for immunohistochemistry (IHC) analysis and n=9 for behavioral analysis was determined to be appropriate for detecting significant differences ($\alpha$=0.05) at a power of 0.8. Sample randomization was not performed as independent variables were age and genotype. Researchers were blinded throughout histological assessments with groups being unblinded at the end of each experiment upon statistical analysis. Data are expressed as mean ± s.e.m with individual sample values being shown. Statistical analysis was performed with Prism 8.0 (GraphPad), R, DESeq2, Seurat, or Monocle 3. Comparisons of means in histology experiments were analyzed with multiple *t*-tests followed by Holm–Sidak correction or mixed effects analysis followed by Dunnett's multiple comparisons (Prism). Changes in expression for aggregated gene sets were determined with one-sample *t*-tests with the expected value of 0 in log2 space (null hypothesis of no change). Differential expression analysis in DESeq2 was based on Wald's significance test of the negative binomial distribution. Differential expression analysis of single-cell RNA-Seq data in Seurat was performed using non-parametric Wilcoxon rank sum tests. Autocorrelation analysis to determine focal expression Monocle 3 was done using Moran's I. The significance of gene set overlap was determined using the $\chi^2$ test with R. All data generated or analyzed in this study are included in this article.

## Results

### Complementary single-cell transcriptional and immunohistochemical analysis reveals the dynamics of heterogeneous hippocampal microglia aging

Previously, single-cell transcriptional profiling of microglia uncovered a diversity of responses during development and in response to pathology aggregated over multiple brain regions (*Hammond et al., 2019*; *Li et al., 2023*; *Keren-Shaul et al., 2017*; *Masuda et al., 2019*; *Li et al., 2019*; *Kracht et al., 2020*). Microglia have region-specific transcriptional states, so we decided to focus our analyses on the hippocampus (*Grabert et al., 2016*), a region with well-reported functional differences during aging (*Fan et al., 2017*). Correspondingly, we investigated age-related transcriptional heterogeneity by performing scRNA-Seq on hippocampal CD11b-positive cells isolated from mice across numerous life stages at mature (6 months), middle-age (12 months), aged (18 months), and old age (24 months) using a commonly employed pooling strategy (*Lee et al., 2023*; *Millet et al., 2024*; *Harrington et al., 2023*) (n=1 pool of five biological replicates for each age) to reveal the dynamics of microglia aging. 82% of cells isolated with CD11b were identified as microglia using canonical

markers with a small subset being proliferative (*Figure 1—figure supplement 1A-C*). Dimensionality reduction revealed that non-proliferative microglia were grouped into interconnected clusters that contained homeostatic, transition, and activated states, along with a semi-distinct cluster with interferon activation (*Figure 1A*, *Figure 1—figure supplement 1D–G*). Overlaying microglia ages onto the clusters revealed progression from a homeostatic state in younger microglia to an activated state in old microglia, suggesting that gradual transcriptional changes occur during microglial aging (*Figure 1B and C*, *Figure 1—figure supplement 1H*). Furthermore, the heterogeneity of microglial transcriptional programs increased with age (*Figure 1—figure supplement 1I*). Expression of *Cx3cr1* and *Itgam* exemplified the homeostatic state, while *B2m*, *Apoe*, *Cd48*, and *Lyz2* depict the activated state (*Figure 1D*). Interestingly, several immediate early genes (*Jun*, *Klf2*, *Fos*), as well as the purinergic receptor gene *P2ry12*, exhibited transient increases in expression at middle-age (*Figure 1D*). Alternatively, the interferon cluster represents a small proportion of the overall microglia population with little age-related changes in the number of these cells, but augmented expression of interferon genes during aging (*Figure 1A, C, and D*, *Figure 1—figure supplement 1G*). Thus, gene expression changes exhibited unique patterns during aging, with certain genes showing age-related decreases (e.g., *Tgfbr1*) or increases (e.g., *Apoe*), while a subset of genes (e.g., *Tgfb1*) have peaks of expression at middle age (*Figure 1D, E, and G*; *Supplementary file 1*). Several complement genes implicated in age-related synaptic loss (*Udeochu et al., 2016*; *Shi et al., 2015*) (e.g., *C1q* and *C3*) demonstrated progressive age-related transcriptional increases, and the increased complement activity in microglia was confirmed by immunohistochemistry (*Figure 1E and F*).

The effects of aging on microglia could be progressive or sudden, so we investigated if we could find evidence of early aging of hippocampal microglia. We observe relatively few differentially genes between 6- and 12-month (*Figure 1G and H*; *Supplementary file 1*). The number of differentially expressed genes gradually increased through the 24-month time point (*Figure 1G–J*; *Supplementary file 1*), suggesting that hippocampal microglia progressively accumulate aged-related expression changes. Furthermore, when investigating the outcome for genes with significant changes for each age, we find that genes that are differentially expressed at 12 months lose their expression change, while genes that change at later ages show progressive expression differences (*Figure 1H*). GO analysis of the microglia from 6- and 12-month-old mice revealed that activation and translational programs are modestly enriched in genes with increased expression during early aging (*Figure 1I*). The latter stages of aging display more prominent enrichment of immune activation and translation processes in genes with increased expression, while revealing that genes that regulate transcriptional processes exhibit decreased expression (*Figure 1J and K*). Thus, microglia progressively age with subsets of cells at each age group being forerunners toward inflammatory activation or refractory to aging by remaining homeostatic. This analysis further suggests that microglia could pass through transitory states during aging.

As scRNA-Seq analysis revealed transcriptional heterogeneity during hippocampal microglial aging, we next assessed microglial phenotypic diversity across hippocampal subregions that could mirror the differential impact of aging across these hippocampal subregions at a functional level (*Kozareva et al., 2019*; *Seki and Arai, 1995*). We subdivided the adult hippocampus into functional units and quantified accumulation of puncta of the lysosomal marker CD68 in microglia to identify activation in the dentate gyrus (DG), molecular layer (ML), granule cell layer (GC), hilus, CA3, and CA1 at 3, 6, 12, 18, and 24 months of age (*Figure 1—figure supplement 2A–C*). We find striking spatial differences in microglial activation patterns during aging, with the GC, hilus, CA3, and outer CA1 regions showing robust microglia activation that increases beginning at middle age, while the ML and inner CA1 exhibit no age-related activation (*Figure 1—figure supplement 2D–F*). Additionally, we characterized the levels of the pro-inflammatory transcription factor NFKB p65 in microglia during aging (*Taniguchi and Karin, 2018*). Microglia-specific expression of NFKB p65 demonstrates region- and age-specific accumulation that mirrors, in part, age-related microglial CD68 increase (*Figure 1—figure supplement 2G–I*). These results indicate temporally defined and spatially heterogeneous microglia activation within the aging hippocampus that complements the identification of heterogeneous transcriptional activation of microglia during aging.

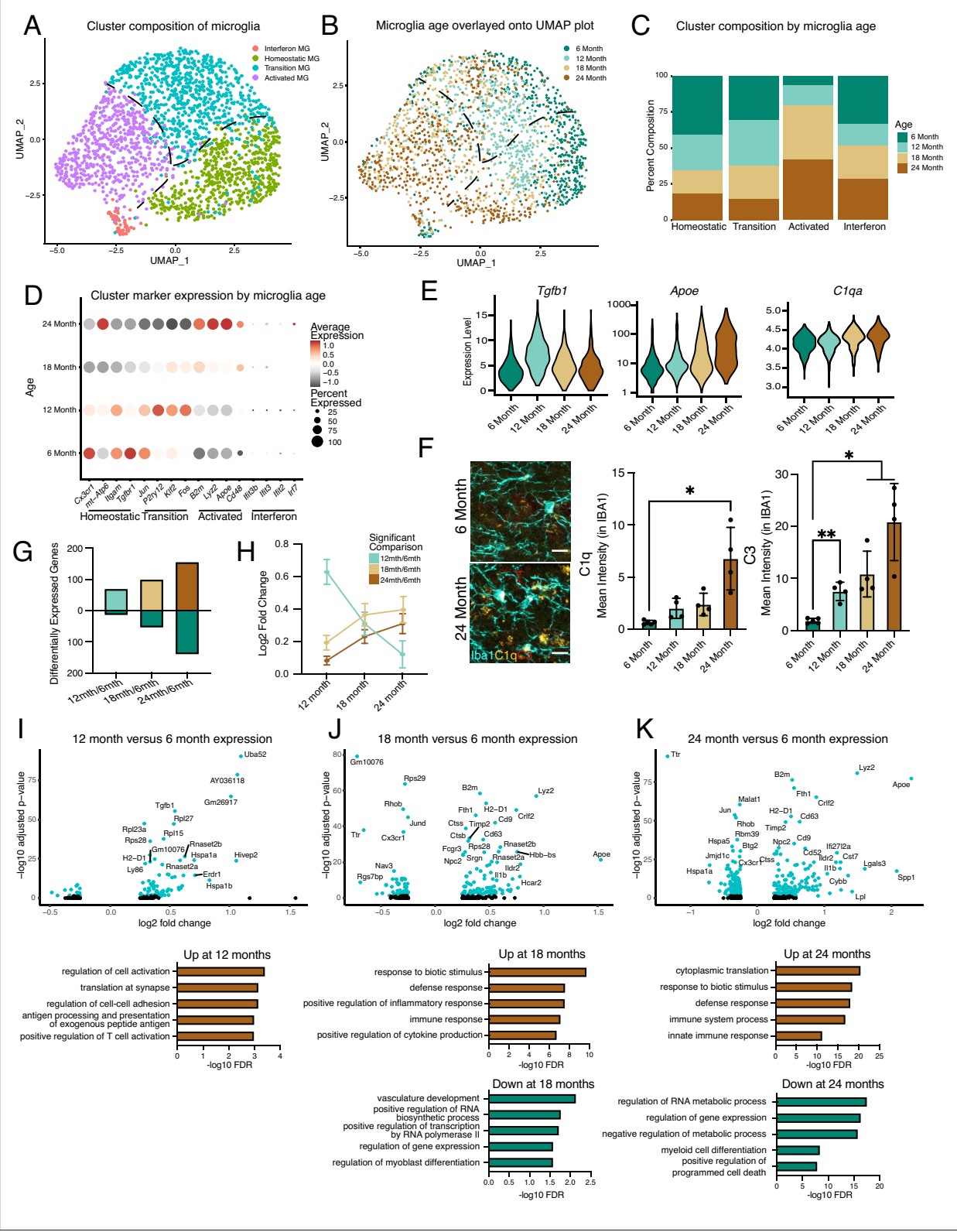

**Figure 1.** Hippocampal microglia exhibit progressive age-related transcriptional inflammatory activation. (**A**) UMAP plot of microglia separated into transcriptional clusters (n=1 pool of five animals for each age). (**B**) Superimposition of ages onto the UMAP plot with dashed lines signifying relative cluster demarcation. (**C**) Percent composition of each cluster by age. (**D**) Dotplot of expression of cluster markers sorted by age. Percent of cells expressing the gene and average normalized expression are represented. (**E**) Violin plots of genes with dynamic age-related expression patterns. (**F**)

*Figure 1 continued on next page*

Figure 1 continued

Representative images and quantification of C1q (yellow), C3 (red), and Iba1 (cyan) staining in the hilus and molecular layer (ML) of 6- and 24-month-old mice (n=4–5 mice per group; mixed effects analysis followed by Dunnett's multiple comparisons; *p<0.05, **p<0.01, ****p<0.0001). (**G**) Number of differentially expressed genes from the 6-month timepoint at each age. Bars above the intersect represent increased expression and those below represent decreased expression. (**H**) The average expression change at all ages for genes differentially regulated genes at individual ages represented by the color scheme in (**B, I, J, K**), Volcano plots of differentially expressed genes in microglia between 6 and 12 months (**I**), 6 and 18 months (**J**), and 6 and 24 months (**K**) and corresponding gene ontology analysis of genes with significantly increased (brown) or decreased (green) expression for each comparison. Data are shown as mean ± s.e.m.

The online version of this article includes the following figure supplement(s) for figure 1:

**Figure supplement 1.** Hippocampal microglia exhibit age-related heterogeneity during aging.

**Figure supplement 2.** Spatiotemporal kinetics of microglial inflammatory activation in the aging hippocampus.

## Trajectory analysis indicates that aging microglia pass through intermediate states

To determine the transcriptional progression of microglia aging toward an activated state, we performed pseudotime analysis of the scRNA-Seq data using Monocle and observed several trajectories (*Figure 2A*). We find that the pseudotime trajectories progressed along the biological ages of the mice, indicating the suitability of this analysis for modeling aging progression in microglia (*Figure 2B*). We focus our analysis on one trajectory with two semi-distinct subbranches that terminated in transcriptional inflammatory activation since the alternate branch had minimal transcriptional differences between the beginning and end of the trajectory (*Figure 2B and C*). To gain insight into the transcriptional states of microglial aging along this inflammatory trajectory, we used spatial autocorrelation analysis and identified five expression modules of coregulated genes (*Figure 2B and C*; *Supplementary file 2*). We named these modules according to the most enriched GO pathways and known roles of prominent genes for microglia. This trajectory proceeds from high expression of homeostatic genes and mitochondrial processes (module 1) to transient expression of stress response genes and *Tgfb1* (module 2), of which TGFβ signaling is critical for microglia development (*Butovsky et al., 2014*; *Bedolla et al., 2024*). A concerted increase in the expression of ribosomal genes (module 3) precedes one subbranch of inflammatory activation (module 4), while the other myeloid activation subbranch (module 5), characterized by *B2m* and *C1qc* expression, proceeds independently of increased ribosomal gene expression. GO analysis of genes specifically upregulated at 12 months showed an enrichment of stress response pathways (*Figure 2—figure supplement 1A and B*), corroborating the findings of the pseudotime analysis that intermediate stages of microglial aging pass through stress response pathways. We complemented Monocle pseudotime analysis using another distinct trajectory analysis (Scorpius) that uncovered similar progression through stress–response and translational intermediate states, followed by inflammatory activation (*Figure 2—figure supplement 1C and D*). We further corroborate the transient nature of the intermediate stress response module (module 2), observing KLF2 expression that peaks at 18 months of age by immunohistochemistry (*Figure 2H*) and *Tgfb1* expression that has a transient peak at 12 months of age by RNAscope (*Figure 3B*). Thus, pseudotime trajectory analysis identifies a progression of intermediate states of microglial aging.

The gene-level dynamics of microglia aging are further revealed when interrogating the modules at each timepoint. Expression of several genes in the homeostatic cluster, including *Cx3cr1* and *Tgfbr1*, progressively decline throughout the aging timeline (*Figure 2D*). Stress–response gene expression peaks at early timepoints during aging, while those genes in the translation module have coherent upregulation at 24 months of age (*Figure 2D*). The top-ranked genes in the inflammation module gradually increase their expression throughout aging in a consistent manner (*Figure 2D*). Alternatively, several genes in the myeloid activation module have expression peaks at 12 (*Cst3, Selplg*) or 18 months (*Ctss, Cd9*) of age (*Figure 2D*). When the modules are collapsed into metagenes, we observe that the homeostatic module is reduced early on in aging, while the stress response module follows later (*Figure 2E–G*). The ribosomal and inflammatory modules display augmented expression at latter ages (*Figure 2E–G*). Alternatively, the myeloid activation consistently has slightly increased expression throughout aging. These results suggest that with advancing age microglia lose homeostatic and stress response functions, while increasing their translational and inflammatory capacity.

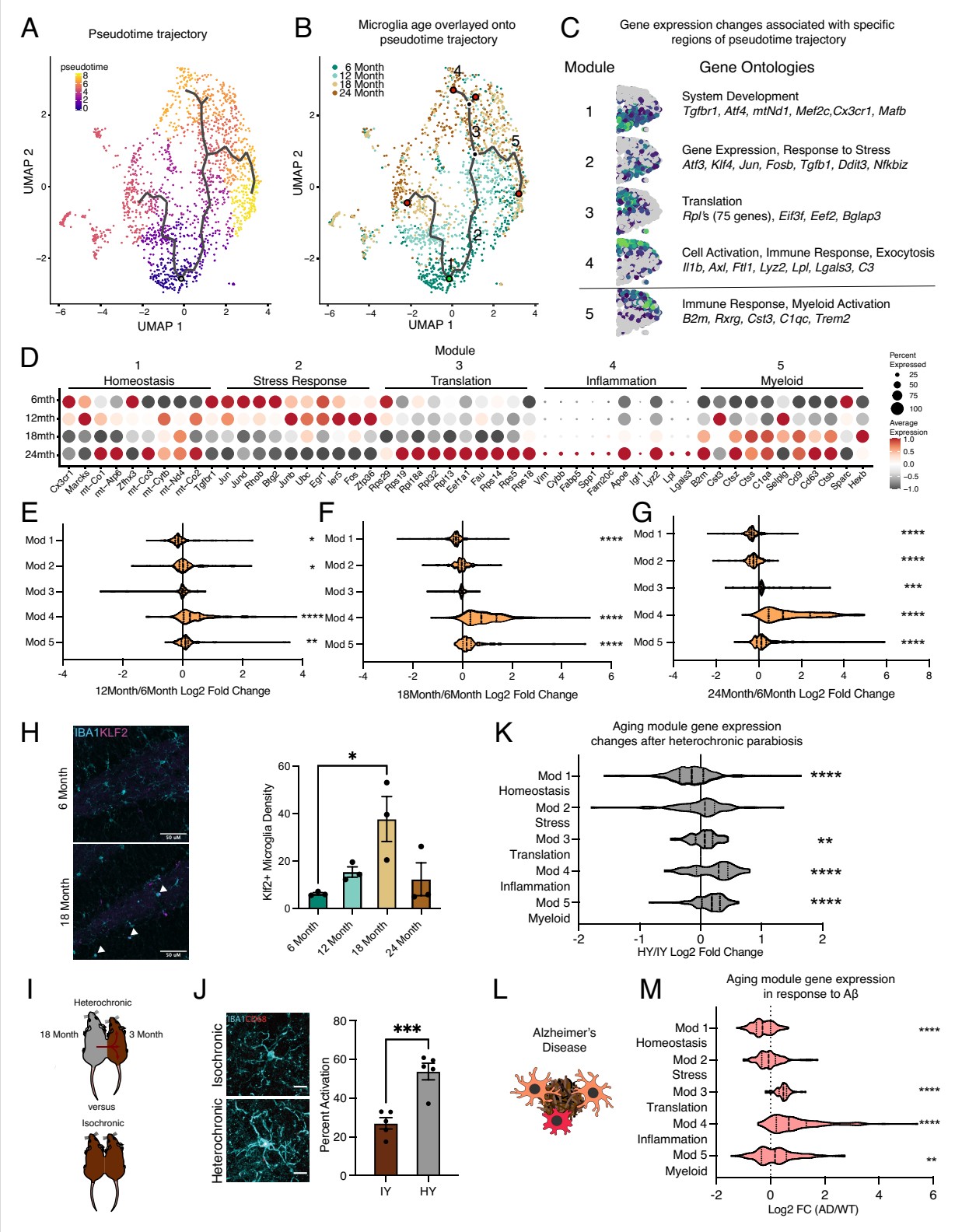

**Figure 2.** Hippocampal microglia aging advances through intermediate states that respond to systemic interventions or disease states. (**A**) Pseudotime trajectories of microglia from an anchor point located in 6-month microglia presented in a UMAP plot (n=1 pool of five animals for each age). (**B**) Microglia ages superimposed over pseudotime trajectories. (**C**) Gene expression modules representing sections of the inflammatory aging trajectory over the right half of the UMAP plot (left). Modules were discovered using Moran's I autocorrelation test. Top gene ontology terms and

*Figure 2 continued on next page*

*Figure 2 continued*

representative genes in each module (right). (**D**) Dotplot of pseudotime modules sorted by age. Percent of cells expressing the gene and average normalized expression are represented. (**E–G**) Average gene expression changes for each aging module represented as log2 fold change of 12 months (**E**), 18 months (**F**), or 24 months (**G**) over 6 months. (**H**) Representative images and quantification of KLF2 (magenta) and IBA1 (cyan) staining in the hippocampus across ages (n=3 mice per group; one-way ANOVA with Tukey's post-hoc test; *p<0.05). (**I**) Diagram of the heterochronic parabiosis model with the comparisons made in scRNA-Seq. (**J**) Representative images and quantification of CD68 (red) and IBA1 (cyan) staining in the hippocampus of isochronic young (IY) and heterochronic young (HY) (n=5 mice per group; unpaired Student's *t*-test; ***p<0.001). (**K**) Average gene expression changes for each aging module represented as log2 fold change of heterochronic young (HY) over isochronic young (IY) adult parabionts. Data from *Pálovics et al., 2022*. (**L**) Diagram of microglia surrounding an Aβ plaque. (**M**) Average gene expression changes for each aging module represented as log2 fold change of the *App^{NL-G-F}* genotype (AD) over wildtype (WT). Data from *Sala Frigerio et al., 2019* (one-sample *t*-test with the expected value of 0 [no change]; *p<0.05, **p<0.01, ***p<0.001, ****p<0.0001). Data are shown as mean ± s.e.m.

The online version of this article includes the following figure supplement(s) for figure 2:

**Figure supplement 1.** Hippocampal microglia aging advances through intermediate states that respond to systemic interventions or disease states.

Next, we used models of pro-aging systemic interventions (*Bieri et al., 2023*) and age-related neurodegenerative disease (*Ransohoff, 2016*) to determine their impact on hippocampal microglia aging trajectories. First, we used the heterochronic parabiosis model (*Pálovics et al., 2022*) to investigate whether exposure to an aged systemic environment could promote progression of young adult microglia along the identified aging trajectory (*Figure 2I*). Performing immunohistochemistry for IBA1 and CD68, we reveal that an aged systemic environment activates microglia (*Figure 2J*). Next, we utilized publicly available scRNA-Seq datasets of parabiosis (*Pálovics et al., 2022*) and Alzheimer's disease (*Sala Frigerio et al., 2019*) models to investigate the transcription consequences of these interventions on microglia aging trajectories. Analyzing the scRNA-Seq dataset from *Pálovics et al., 2022*, we find that hippocampal microglia from young adult heterochronic parabionts have decreased expression of homeostatic genes (module 1), while having increased expression of ribosomal genes (modules 3), inflammatory activation genes (module 4), and myeloid activation genes (module 5) (*Figure 2K*, *Figure 2—figure supplement 1E*). Using scRNA-Seq from Sala Frigerio et al., we analyzed the *App^{NL-G-F}* transgenic mouse model data at 12 months of age to determine the effects of Alzheimer's disease pathology on microglial aging (*Sala Frigerio et al., 2019*; *Figure 2L*). We observe exaggerated advancement along the aging trajectory in *App^{NL-G-F}* mice compared to controls, as expression is shifted toward the age-associated modules (*Figure 2M*, *Figure 2—figure supplement 1F and G*). These shifts in gene expression posit that the aged systemic and diseased environments drive adult microglia advancement along an aging-associated transcriptional trajectory.

## Intermediate states of microglia aging act as modulators of age-related trajectory progression

Next, we sought to further corroborate microglial age-related molecular changes associated with individual intermediate states in the aging hippocampus. Specifically, we assessed changes in the intermediate stress response (module 2) and translation (module 3) modules by examining expression of *Tgfb1* (*Figure 3A and B*) and the ribosomal protein S6 (*Figure 3G and H*), respectively. The vast majority (>95%) of *Tgfb1* signal localized to microglia at every age (*Figure 3—figure supplement 1A*), indicating microglia are the predominant source for hippocampal TGFB1. Consistent with scRNA-Seq analysis (module 2) (*Figure 1E*), we observed highest *Tgfb1* expression in microglia by middle-age using RNAscope (*Figure 3B*). TGFBR1 activation was reduced during aging in hippocampal microglia (*Figure 3—figure supplement 1B*), and the genes involved in the TGFβ signaling pathway exhibited age-related expression changes with several having peak expression at 12 months of age (*Figure 2C*, *Figure 3—figure supplement 1C*). Next, we examined at a transcriptional level whether TGFβ1 is likely to act in an autocrine or paracrine fashion in microglia. To do so, we interrogated our scRNA-Seq dataset, leveraging the variation in microglia *Tgfb1* expression to probe the relative activity of TGFβ1. High expression of downstream TGFβ signaling pathway components in microglia with high *Tgfb1* expression would point to autocrine mechanisms while, alternatively, high expression of downstream TGFβ signaling pathway components in microglia with low *Tgfb1* expression would point to paracrine mechanisms. We observed highest expression of TGFβ signaling pathway components and targets in microglia with the highest expression of *Tgfb1* (*Figure 3—figure supplement 1E–G*), suggesting an autocrine mechanism of action. Additionally, consistent with an increase in translational components

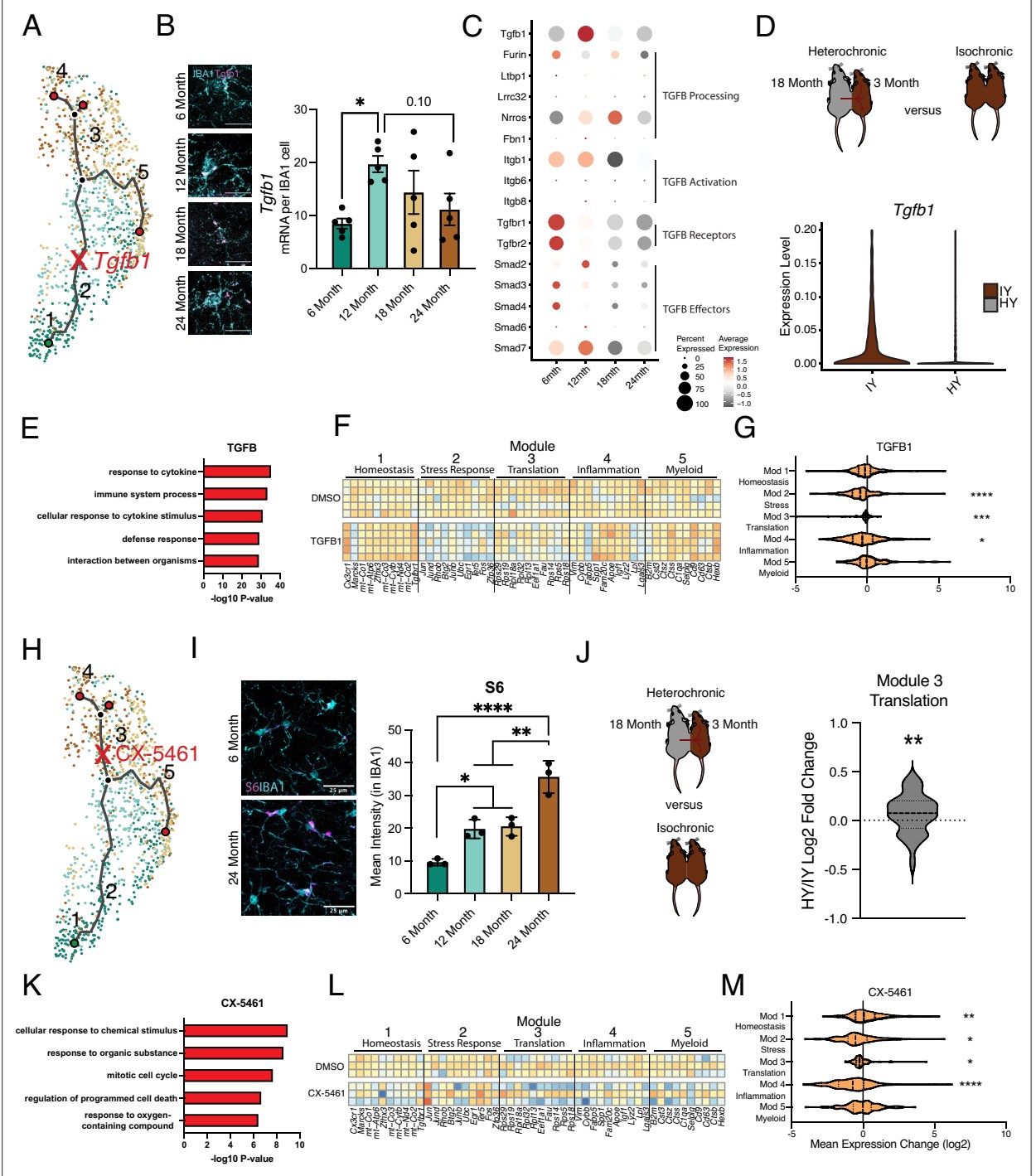

**Figure 3.** Intermediate states of microglia aging act as checkpoints on inflammatory progression. (**A**) Representation of the microglia aging trajectory over the UMAP plot highlighting the region of peak *Tgfb1* expression. (**B**) Representative RNAscope images and quantification of *Tgfb1* (red) expression in IBA1 (cyan) cells across ages (n=5 per group; one-way ANOVA with Dunnett's post hoc test; *p<0.05). (**C**) Dotplot of the expression values of TGFB1 signaling components from scRNA-Seq of aging hippocampal microglia (6-, 12-, 18-, and 24-month-old). Percent of cells expressing the gene and average normalized expression are represented. (**D**) Schematic of the heterochronic parabiosis model and quantification of hippocampal microglia expression of *Tgfb1* from isochronic young (IY) and heterochronic young (HY) adult parabionts. Data derived from *Pálovics et al., 2022*. (**E**) Top gene ontology terms for the set of genes with significantly decreased expression in bulk microglia RNA-Seq following TGFB1 treatment compared to control (DMSO) in LPS-treated microglia (n=5 per group). (**F**) Heatmap of top 10 genes in each aging module following TGFB1 compared to DMSO in LPS-treated microglia. (**G**) Average gene expression changes for each aging module represented as log2 fold change of TGFB1 treatment over DMSO (one-sample *t*-test with the expected value of 0 [no change]; *p<0.05, ***p<0.001, ****p<0.0001). (**H**) Representation of the microglia aging trajectory over

*Figure 3 continued on next page*

Figure 3 continued

the UMAP plot highlighting the stage where CX-5461 modulates the trajectory. (**I**) Representative images of S6 (magenta) and Iba1 (cyan) staining in the hippocampus of 6- and 24-month-old mice and quantification across aging (n=3 mice per group; one-way ANOVA with Tukey's post hoc test; *p<0.05, **p<0.005, ****p<0.0001). (**J**) Schematic of the heterochronic parabiosis model and quantification of hippocampal microglia expression of translation module from isochronic young (IY) and heterochronic young (HY) adult parabionts. Data derived from *Pálovics et al., 2022* (one-sample *t*-test with the expected value of 0 [no change]; **p<0.01). (**K**) Top gene ontology terms for the set of genes with significantly decreased expression in bulk microglia RNA-Seq following CX-5461 treatment compared to control (DMSO) in LPS-treated microglia (n=3 per group) (**L**) Heatmap of top 10 genes in each aging module following CX-5461 compared to DMSO in LPS-treated microglia. (**M**) Average gene expression changes for each aging module represented as log2 fold change of CX-5461 treatment over DMSO (one-sample *t*-test with the expected value of 0 [no change]; *p<0.05, **p<0.01, ****p<0.0001).

The online version of this article includes the following figure supplement(s) for figure 3:

**Figure supplement 1.** Intermediate states of microglia aging act as checkpoints on inflammatory progression.

found during the scRNA-Seq analysis (module 3), we observed increased expression of ribosomal protein S6 (*Figure 3I*; *Yi et al., 2021*) by immunohistochemistry.

To complement hippocampal aging analysis, we assessed the impact of the aging systemic milieu on both the TGFβ pathway (module 2) and translational components (module 3) in the young adult hippocampus following heterochronic parabiosis. Using the dataset from *Pálovics et al., 2022*, we observed a decrease in microglia *Tgfb1* and TGFβ pathway expression in young adult heterochronic parabionts compared to age-matched young adult isochronic controls (*Figure 3C*, *Figure 3—figure supplement 1D*). Additionally, expression of the translation module in microglia increased following exposure to an aged systemic environment in the heterochronic parabiosis model (*Figure 3J*).

To investigate the role of the intermediate stress response (module 2) and translation (module 3) modules in mediating advancement through microglial activation states, we used an in vitro approach. Primary microglia were treated with LPS, which induced gene expression changes with a significant overlap to aging, as well as expression changes in aging modules genes (*Figure 3—figure supplement 1H and I*; *Supplementary file 3*). We administered TGFB1 to activate TGFβ signaling (module 2) (*Figure 3*) and CX-5461 to inhibit RNA Pol I synthesis and interfere with translation (module 3) (*Figure 3H*) in the context of microglial activation. TGFB1 treatment modified the transcriptional states of LPS-treated microglia (*Figure 3—figure supplement 1J*) and caused decreased expression of genes in GO terms related to inflammatory immune processes after LPS stimulation (*Figure 3E*). TGFB1 treatment reduced expression of later aging modules (modules 2, 3, and 4) and restored expression of the homeostatic gene *Cx3cr1* (*Figure 3F and G*), suggesting that TGFβ signaling acts as an aging modulator during microglial stress response. Targeting translation (module 3) with CX-5461 attenuated expression of genes in GO terms related to immune processes after LPS stimulation (*Figure 3K*; *Supplementary file 3*), as well as decreased expression of genes in modules 2, 3, and 4, while restoring expression of homeostatic genes in module 1 (*Figure 3*). Interestingly, CX-5461 treatment had no effect on genes in module 5, which is located on an independent myeloid activation trajectory from altered translational gene expression (*Figure 3L and M*). These in vitro perturbation data identify active roles for intermediate states in mediating advancement along aging-associated inflammatory trajectories.

## Mimicking age-related changes in microglia-derived TGFB1 promotes microglial advancement along an aging-associated inflammatory trajectory in vivo

Having uncovered roles for intermediate states in advancing microglia along aging-associated trajectories in vitro, we next investigated the role of individual aging modules in adult microglia in vivo. We elected to disrupt the transition from homeostasis to inflammatory activation by targeting microglia-derived TGFB1, a key marker of the stress response intermediate state (module 2). While TGFB1 is critical for microglia development (*Bedolla et al., 2024*; *Spittau et al., 2020*), the role of microglia-derived TGFB1 in regulating microglia homeostasis during aging remains to be defined.

We generated *Tgfb1*<sup>flox/flox</sup>, *Tgfb1*<sup>flox/wt</sup>, and *Tgfb1*<sup>wt/wt</sup> mice carrying an inducible *Cx3cr1*<sup>Cre-ER</sup> gene, in which *Tgfb1* is excised specifically in mature (7–8 months) microglia upon tamoxifen administration (*Tgfb1* cKO, *Tgfb1* Het, and WT, respectively) (*Figure 4A*). To disrupt the age-related increase in microglia *Tgfb1* observed between mature and middle-age (*Figures 1E and 3B*), mature mice were

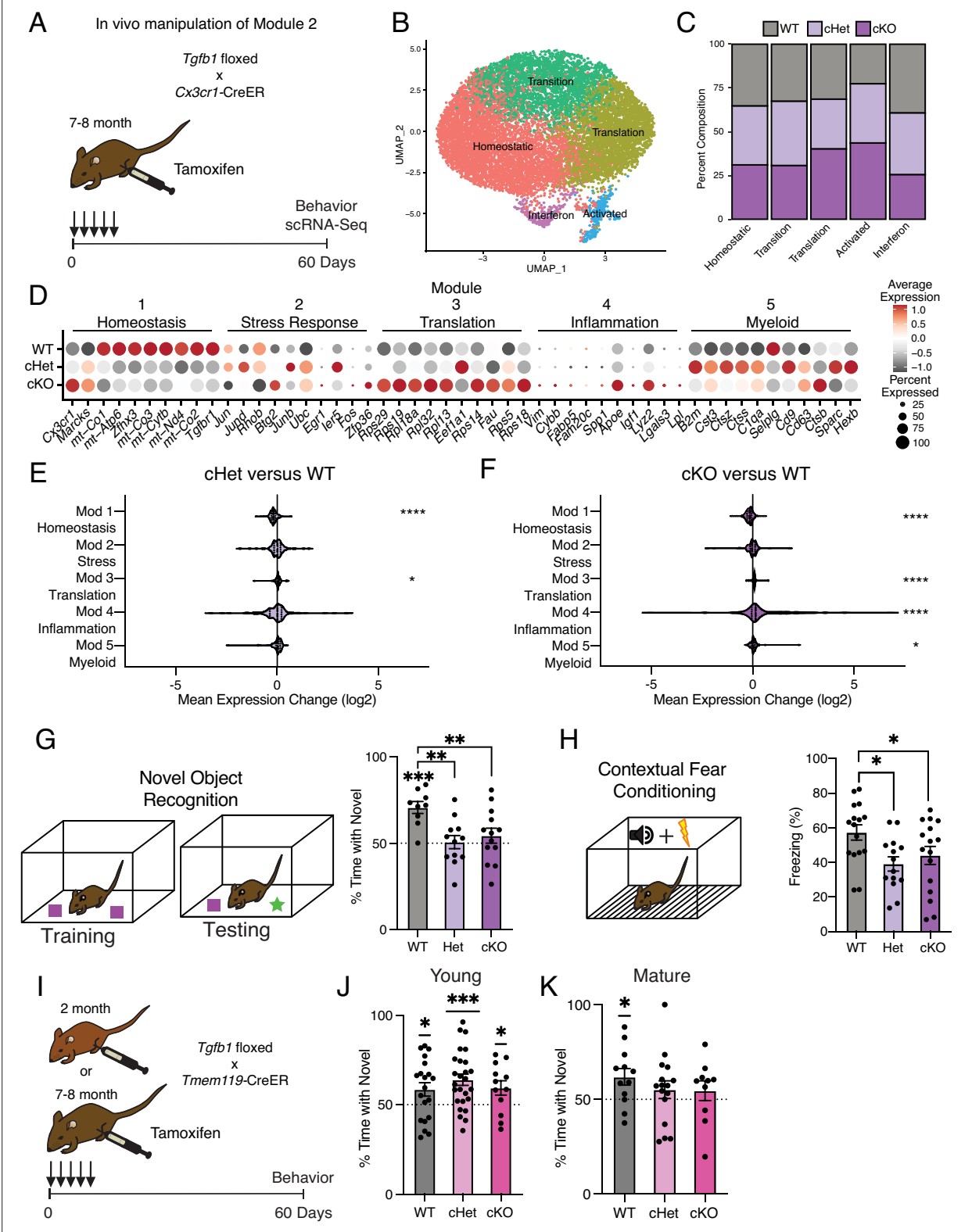

**Figure 4.** Targeting age-related changes in microglia-derived TGFB1 promotes microglia advancement along inflammatory trajectories in the hippocampus and impairs cognition. (**A**) Schematic of in vivo manipulation schema of Module 2. Mature (7–8 months) littermate *Cx3cr1*<sup>Cre-ER/+</sup>; *Tgfb1*<sup>flox/flox</sup> (cKO), *Cx3cr1*<sup>Cre-ER/+</sup>; *Tgfb1*<sup>flow/wt</sup> (Het), and *Cx3cr1*<sup>Cre-ER/+</sup>; *Tgfb1*<sup>wt/wt</sup> (WT) mice were administered tamoxifen and subject to hippocampal microglia scRNA-Seq and behavior analysis two months later. (**B**) UMAP plot of microglia separated into transcriptional clusters (n=2 pools of three animals per

*Figure 4 continued on next page*

*Figure 4 continued*

genotype). (**C**) Stacked bar plot of the normalized relative percentage of cells of each genotype in the identified clusters. (**D**) Dotplot of expression of aging module markers sorted by genotype. Percent of cells expressing the gene and average normalized expression are represented. (**E, F**) Average gene expression changes for each aging module represented as log2 fold change of either Het over WT (**E**) or KO over WT (**F**) (one-sample *t*-test with the expected value of 0 [no change]; *p<0.05, ****p<0.0001). (**G**) Novel object recognition task. (Top) Diagram of the training and testing phases of the novel object recognition paradigm. (Bottom) Quantification of the NOR testing phase represented as a percentage of time spent with the novel object (over the total time spent interacting with the objects) (n=9–13 per genotype; one-sample *t*-test with the expected value of 50 [equal time spent with each object]; ***p<0.001) (differences between groups determined by one-way ANOVA; **p<0.01). (**H**) Contextual fear conditioning. (Top) Diagram of the fear conditioning paradigm. (Bottom) Quantification of the percentage of time mice froze in the contextual fear conditioning testing phase (n=14–16 per genotype; one-way ANOVA; *p<0.05). Data are shown as mean ± s.e.m. (**I**) Schematic of in vivo manipulation of *Tgfb1* at young and mature ages. Young (2 month) or mature (7–8 months) littermate *Tmem119$^{Cre-ER/+}$; Tgfb1$^{flox/flox}$* (cKO), *Tmem119$^{Cre-ER/+}$; Tgfb1$^{flow/wt}$* (Het), and *Tmem119$^{Cre-ER/+}$; Tgfb1$^{wt/wt}$* (WT) mice were administered tamoxifen and subject to behavioral analysis two months later. (**J, K**) Quantification of the NOR testing phase represented as a percentage of time spent with the novel object (over the total time spent interacting with the objects) for young (**J**) and mature (**K**) *Tmem119$^{Cre-ER}$::Tgfb1* mice n=12–26 in young and n=10–16 in mature mice per genotype; one-sample *t*-test with the expected value of 50 (equal time spent with each object; *p<0.05, ***p<0.001).

The online version of this article includes the following figure supplement(s) for figure 4:

**Figure supplement 1.** Targeting age-related changes in microglia-derived TGFB1 promotes microglia advancement along inflammatory trajectories in the hippocampus.

**Figure supplement 2.** Targeting age-related changes in microglia-derived TGFB1 impairs cognition.

administered tamoxifen and molecular changes were assessed two months later (*Figure 4A*). We find that microglia activation, as measured by CD68/IBA1 immunohistochemistry, exhibited genotype-dependent effects (*Figure 4—figure supplement 1A*). We performed scRNA-Seq on microglia isolated from WT, *Tgfb1* Het, and *Tgfb1* cKO mature mice (n=2 pools of three animals per genotype) and detected genotype-dependent changes in *Tgfb1*expression levels, as well as effectors of TGFB1 signaling (*Figure 4—figure supplement 1B*). Dimensionality reduction and clustering of cells from scRNA-Seq revealed readily identifiable clusters based on marker expression (*Figure 4B*). Composition of these clusters were dependent on the microglia genotype, with WT microglia representing the largest fraction of the homeostatic cluster and *Tgfb1* cKO microglia representing the largest fraction of the activated cluster (*Figure 4C*, *Figure 4—figure supplement 1C*). Next, we assessed whether deletion of *Tgfb1* in mature adult microglia impacted advancement along aging-associated inflammatory trajectories. Interestingly, when we analyze the effects of *Tgfb1* dosage on the aging modules, we find that the loss of a single *Tgfb1* allele reduces expression of homeostatic genes (module 1), while increasing expression of translation-related genes (module 3) (*Figure 4D and E*). In addition to a decrease in the expression of homeostatic genes (module 1), loss of both *Tgfb1* alleles caused a greater increase in the expression of translation-related genes (module 3) and led to a profound increase in inflammatory activation genes (module 4) (*Figure 4D and F*). Results were corroborated in an independent cohort of mature *Tgfb1* cKO mice, in which we detected altered levels of immune- and age-related cell surface markers (*Figure 4—figure supplement 1D and E*), transcriptional alterations that significantly overlapped with aging (*Figure 4—figure supplement 1F–H*; *Supplementary file 4*), and progression along the aging-associated inflammatory trajectories (*Figure 4—figure supplement 1*). These data indicate that microglia-derived TGFB1 is both necessary for expression of youth-associated homeostatic genes and prevents aberrant microglia inflammatory activation, further suggesting that TGFBβ signaling acts as a modulator in mediating advancement along aging-associated inflammatory trajectories.

## Mimicking age-related changes in microglia-derived TGFB1 impairs hippocampal-dependent cognitive function

It is becoming increasingly apparent that microglia interact with other cell types in the brain to influence cognitive function (*Salter and Beggs, 2014*; *Cserép et al., 2021*). Given that manipulating the levels of microglia-derived TGFB1 altered microglia homeostasis and progression along aging-associated trajectories, we next investigated the functional consequence of the loss of microglia-derived TGFB1 on cognition. We assessed hippocampal-dependent learning and memory in WT, *Tgfb1* Het, and *Tgfb1* cKO mature mice using novel object recognition and contextual fear conditioning – behavioral paradigms that are sensitive to age-related impairments (*Horowitz et al., 2020*;

*Traschütz et al., 2018*). During novel object recognition testing, WT mature mice were biased toward a novel object relative to a familiar object while neither *Tgfb1* Het nor *Tgfb1* cKO mature mice showed any preference (*Figure 4G*). During contextual fear conditioning testing, *Tgfb1* Het and *Tgfb1* cKO mature mice exhibited less freezing compared to WT controls (*Figure 4H*). Alternatively, we assessed amygdala-dependent fear memory by cued fear conditioning (*Figure 4—figure supplement 2A*; *Phillips and LeDoux, 1992*) and short-term spatial reference memory by Y maze (*Figure 4—figure supplement 2B*; *Kraeuter et al., 1916*) and observed no differences across genotypes. As a control, we also profiled general health using an open-field paradigm and observed no differences in total distance traveled or time spent in the center of the open field, indicative of normal motor and anxiety functions (*Figure 4—figure supplement 2C*).

To further examine the age-dependent role of microglia-derived TGFB1 in cognitive function and complement our genetic approach, we utilized an alternative inducible microglia-specific *Tmem119*-CreER mouse model (*Kaiser and Feng, 2019*) to manipulate *Tgfb1* expression at both young (2–3 months) and mature (7–8 months) ages (*Figure 4I*). We assessed hippocampal-dependent learning and memory in WT, *Tgfb1* Het, and *Tgfb1* cKO young and mature mice using novel object recognition. During testing, young mice were biased toward a novel object relative to a familiar object regardless of genotype (*Figure 4J*), indicating normal cognitive function despite loss of microglia-specific *Tgfb1* at young age. However, while WT mature mice were biased toward a novel object relative to a familiar object, both *Tgfb1* Het or *Tgfb1* cKO mature mice lost their preference for the novel object (*Figure 4K*), indicating an age-dependent role for microglia-specific *Tgfb1* in maintaining cognitive function. No differences were observed in either Y maze (*Figure 4—figure supplement 2D and F*) or open field (*Figure 4—figure supplement 2E and G*) between genotypes at either young or mature ages. These behavioral data indicate that loss of microglia-derived TGFB1 not only impacts aging-associated trajectories but also promotes hippocampal-dependent cognitive impairments characteristic of age-related functional decline in an age-dependent manner.

## Discussion

Cumulatively, we demonstrate that microglia aging in the hippocampus advances through intermediate states that precede inflammatory activation, with functional implications for age-related cognitive decline. These transient states, such as activation of the stress response and TGFβ signaling, act as modulators of further age-related activation. Microglia demonstrate spatiotemporal dissimilarities in advancement along aging trajectories across the hippocampus. Furthermore, microglia progression through this aging trajectory is in turn influenced by the systemic environment and neurodegenerative disease states.

Analysis of our scRNA-Seq data elucidates cryptic, but necessary, intermediate states of microglia progression to age-related inflammatory activation. These states are present in middle-aged mice and constitute stress– response pathways likely responsive to internal and external cellular insults (*Wek, 2018*; *López-Otín et al., 2013*). Pseudotime analysis suggests a highly plausible sequence of molecular alterations that lead to age-related inflammatory activation of hippocampal microglia. First, mitochondrial dysfunction triggers a stress response, and products of this stress response (e.g., TGFB1) attempt to return microglia to homeostasis (*Chovatiya and Medzhitov, 2014*). Here, chronic activation of the stress response leads to subsequent increased translational capacity that promotes inflammatory activation (*Xu et al., 2020*). Alternatively, another branch of microglial activation advances independently of increased translational capacity, evidenced by translational inhibition having little effect on its activation following LPS stimulation. Thus, we find that increases in translational capacity predicate age-related inflammatory phenotypes, identifying potential targets to counter microglial aging. Analysis of scRNA-Seq data further identifies intermediate states of microglial aging in young adult heterochronic parabionts, suggesting a pivotal role for the aged systemic environment in driving advancement toward age-related inflammatory activation.

While we find evidence of the beginnings of inflammatory activation in the hippocampus by middle age (12 months), consistent with the timing recently reported in the whole brain (*Li et al., 2023*), it should be noted that scRNA-Seq analysis of the whole brain did not observe intermediate stages in microglial aging (*Li et al., 2023*). These disparate observations could likely be the result of analyzing multiple regions with highly differentiated transcriptional programs (*Grabert et al., 2016*) that could obfuscate regional aging trajectories. This further underscores the need for both temporal and regional

specificity in aging studies, given the level of cellular heterogeneity observed across ages and brain regions. Indeed, in the hippocampus, it is likely that the local microenvironment plays a significant role in the genesis of microglial aging, as regional distinctions in cellular function and composition – even in apposed subregions – elicit different outcomes in microglia with age and following exposure to an aged systemic environment. With age, multiple hippocampal subregions accumulate activated microglia beginning at 12 months of age, signifying that age-related alterations in microglia are initially present at middle age. Studying microglia, or other cell types, at these early stages of aging in a region-specific manner could represent useful models to investigate the ontogeny of aging without the confounding effects of accumulated functional deficits. Alternatively, the lack of age-related accumulation of activated microglia in the molecular layer and inner CA1, hippocampal subregions rich in synaptic terminals, points toward microenvironmental influences that may prove refractory to aging.

Of note, we find that an inflammatory trajectory of microglia aging can be mitigated in vitro by environmental TGFβ signaling, while loss of microglia-derived TGFB1 promoted progression along aging-associated inflammatory trajectories in vivo. Together, these data indicate that TGFβ signaling is necessary for maintaining youth-associated homeostasis and represents a modulator in mediating advancement along aging-associated inflammatory trajectories. Furthermore, our results, combined with those of *Bedolla et al., 2024*, strongly suggest that TGFβ signals in an autocrine fashion to return microglia to a homeostatic state after encountering stressful stimuli. Functionally, loss of microglia-derived TGFB1 resulted in hippocampal-dependent cognitive impairments in an age-dependent manner, suggesting that advancement through aging-associated trajectories may facilitate age-related cognitive decline. Given that a transient increase in *Tgfb1* expression is observed in middle-aged microglia, it is possible that maintaining stress response-mediated signaling into older ages may also provide a means to delay age-related microglial dysfunction and maintain cognitive function. Alternatively, TGFβ signaling impairs microglial responses to Alzheimer's disease pathology (*Yin et al., 2023*), suggesting that the dynamic regulation of TGFβ signaling in response to aging-associated pathologies is necessary for tuning microglial responses.

While microglia have been demonstrated to mediate the establishment of cognitive function during development (*Salter and Beggs, 2014*) and play prominent roles in neurodegeneration (*Sarlus and Heneka, 2017*), little is known about how age-related changes in microglia affect cognitive function. The specificity of the cognitive deficits that we observe in *Tgfb1* Het and *Tgfb1* cKO mice suggests that age-related changes in microglia-derived TGFB1 predominantly impact hippocampal-dependent cognitive decline. Furthermore, the deficits observed in *Tgfb1* heterozygous mice point to the importance of maintaining youthful levels of microglia-derived TGFB1 to counteract stressors that accrue during aging.

Altogether, our results highlight the need to investigate aging across the lifespan in a region-specific manner to reveal necessary cellular intermediates in the aging process. While the present study focuses on age-related changes in microglia, our findings have broad implications for other studies in determining the extent to which aging sequelae are conserved across different cell types in the hippocampus. Furthermore, therapeutic interventions targeting intermediate states in the aging process hold far-reaching potential to counteract the development of age-related pathologies, including those promoting cognitive dysfunction.

## Acknowledgements

We thank Dr. Erik Ullian for critically reading manuscript. This work was funded by the National Institute on Aging (AG055292 (JMS), AG055797 (SAV), AG077816 (SAV)), Simons Foundation (SAV). The authors declare no competing financial interests.

## Additional information

### Funding

| Funder | Grant reference number | Author |
| --- | --- | --- |
| National Institute on Aging | AG055292 | Jeremy M Shea |

| Funder | Grant reference number | Author |
|---|---|---|
| National Institute on Aging | AG055797 | Saul A Villeda |
| National Institute on Aging | AG077816 | Saul A Villeda |
| Simons Foundation | | Saul A Villeda |

The funders had no role in study design, data collection and interpretation, or the decision to submit the work for publication.

### Author contributions
Jeremy M Shea, Conceptualization, Resources, Data curation, Formal analysis, Funding acquisition, Validation, Investigation, Visualization, Methodology, Writing - original draft, Project administration, Writing - review and editing; Saul A Villeda, Conceptualization, Supervision, Funding acquisition, Writing - original draft, Project administration, Writing - review and editing

### Author ORCIDs
Jeremy M Shea (iD) https://orcid.org/0000-0003-1965-543X
Saul A Villeda (iD) https://orcid.org/0000-0002-2726-9582

### Ethics
All animal procedures were performed in accordance with protocols approved by the UCSF Institutional Animal Care and Use Committee (IACUC)(AN194088). Animals were housed in SPF barrier facilities and provided continuous food and water along with environmental enrichment. Surgery was performed under isofluorane, and every effort was made to minimize suffering.

Reviewer #2 (Public review): https://doi.org/10.7554/eLife.97671.3.sa1
Author response https://doi.org/10.7554/eLife.97671.3.sa2

# Additional files

### Supplementary files
MDAR checklist

Supplementary file 1. Gene markers for each age group and differential expression between the 6- and 12-, 18-, or 24-month-old microglia in single-cell RNA-Seq of aging hippocampal microglia. The tables contain genes selectively increased or decreased in each age group when compared to every other age group combined and genes differentially expressed between 6- and 12-, 18-, or 24-month microglia. The tables contain the average log fold change, the percentage of cells in each group with expression, and the adjusted p-value.

Supplementary file 2. Pseudotime trajectories of aging microglia. The pseudotime analysis contains the spatial autocorrelation analysis. Moran's I was used to detect focal expression of genes that were consequently constructed into co-regulated modules using Louvain community analysis. The table contains the adjusted p-value (q-value) and module information.

Supplementary file 3. RNA-Seq analysis of primary microglia treated with LPS and TGFB1 or CX-5461. The tables contain the RNA-Seq differential expression analysis of control and LPS activated primary microglia, DMSO and CX-5461-treated primary microglia activated by LPS, and DMSO and TGFB1-treated primary microglia activated by LPS, respectively. The tables contain expression values of genes for the samples that were compared along with mean expression values for all samples combined, log fold change between sample groups, and adjusted p-values. in the RNA-Seq differential expression analysis of control and LPS activated primary microglia, DMSO and CX-5461-treated primary microglia activated by LPS, and DMSO and TGFB1-treated primary microglia activated by LPS, respectively.

Supplementary file 4. Differential expression analysis of control and Tgfb1 cKO hippocampal microglia. The table contains expression values of genes for the samples that were compared along with mean expression values for all samples combined, log fold change between sample groups, and adjusted p-values.

## Data availability

All RNA-Seq and scRNA-Seq data have been deposited in the Gene Expression Omnibus and are publicly available as of the date of publication. The following datasets were generated for this manuscript and deposited in Gene Expression Omnibus: aging single-cell RNA-Seq (GSE179358), in vitro treated primary microglia (GSE179611), Tgfb1 cKO microglia RNA-Seq (GSE190007), and Tgfb1 cKO microglia single-cell RNA-Seq (GSE211340).

The following datasets were generated:

| Author(s) | Year | Dataset title | Dataset URL | Database and Identifier |
|---|---|---|---|---|
| Shea J, Villeda S | 2024 | Microglia aging progresses through functional intermediate steps | https://www.ncbi.nlm.nih.gov/geo/query/acc.cgi?acc=GSE179358 | NCBI Gene Expression Omnibus, GSE179358 |
| Shea J, Villeda S | 2024 | Microglia aging progresses through functional intermediate steps II | https://www.ncbi.nlm.nih.gov/geo/query/acc.cgi?acc=GSE179611 | NCBI Gene Expression Omnibus, GSE179611 |
| Shea J, Villeda S | 2024 | Microglia aging progresses through functional intermediate steps | https://www.ncbi.nlm.nih.gov/geo/query/acc.cgi?acc=GSE190007 | NCBI Gene Expression Omnibus, GSE190007 |
| Shea J, Villeda S | 2024 | Microglia aging advances through intermediate states that drive inflammatory activation and cognitive decline | https://www.ncbi.nlm.nih.gov/geo/query/acc.cgi?acc=GSE211340 | NCBI Gene Expression Omnibus, GSE211340 |

The following previously published datasets were used:

| Author(s) | Year | Dataset title | Dataset URL | Database and Identifier |
|---|---|---|---|---|
| Palovics R, Schaum N, Pisco A, Quake S, Wyss-Coray T | 2022 | Molecular hallmarks of heterochronic parabiosis at single cell resolution | https://www.ncbi.nlm.nih.gov/geo/query/acc.cgi?acc=GSE193093 | NCBI Gene Expression Omnibus, GSE193093 |
| Frigerio CS, Wolfs L, Fattorelli N, Thrupp N, Voytyuk I, Schmidt I, Mancuso R, Chen W, Woodbury M, Srivastava G, Möller T, Hudry E, Das S, Saido T, Karran E, Hyman B, Perry VH, Fiers M, De Strooper B | 2019 | The major risk factors for Alzheimer's disease: Age, Sex and Genes, modulate the microglia response to Aβ plaques (KW) | https://www.ncbi.nlm.nih.gov/geo/query/acc.cgi?acc=GSE127892 | NCBI Gene Expression Omnibus, GSE127892 |

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
