## [Editor Report · eLife Assessment]

This **important** work advances our understanding of the aging trajectory and heterogeneity of hippocampal microglia. The authors provide an in-depth characterization of microglia in young and old mice as well as at intermediate time points, which reveals the existence of intermediate states characterized by a distinct transcriptional signature. The experimental approach is **solid**, especially with the validation of scRNA-seq findings with other methods. The study should be of interest to neuroimmunologists and biologists interested in aging.

---

## [Referee Report · Reviewer #2 (Public review)]

Summary:

The goal of the paper was to trace the transitions hippocampal microglia undergo along aging. ScRNA-seq analysis allowed the authors to predict a trajectory and hypothesize about possible molecular checkpoints, which keep the pace of microglial aging. E.g. TGF1b was predicted as a molecule slowing down the microglial aging path and indeed, loss of TGF1 in microglia led to premature microglia aging, which was associated with premature loss of cognitive ability. The authors also used the parabiosis model to show how peripheral, blood-derived signals from the old organism can "push" microglia forward on the aging path.

Strengths:

A major strength and uniqueness of this work is the in-depth single-cell dataset, which may be a useful resource for the community, as well as the data showing what happens to young microglia in heterochronic parabiosis setting and upon loss of TGFb in their environment.

Weaknesses:

All weaknesses were addressed during revision.

Overall:

In general, I think the authors did a good job following the initial observations and devised clever ways to test the emerging hypotheses. The resulting data are an important addition to what we know about microglial aging and can be fruitfully used by other researchers, e.g. those working on microglia in a disease context.

Comments on revisions:

All my comments were addressed.

---

## [Author Response]

The following is the authors’ response to the original reviews

**Reviewer #1:**
To gain further insight into the dynamics of microglial aging in the hippocampus, the authors used a bioinformatics method known as "pseudotime" or "trajectory inference" to understand how cells may progress through different functional states, as defined by cellular transcriptome (15,16). These bioinformatics approaches can reveal key patterns in scRNAseq / snRNAseq datasets and, in the present study, the authors conclude that a "stress response" module characterized by expression of TGFb1 represents a key "checkpoint" in microglial aging in midlife, after which the cells can move along distinct transcriptional trajectories as aging progresses. This is an intriguing possibility. However, pseudotime analyses need to be validated via additional bioinformatics as well as follow-up experiments. Indeed, Heumos et al, in their Nature Genetics "Expert Guidelines" Review, emphasize that "inferred trajectories might not necessarily have biological meaning." They recommend that "when the expected topology is unknown, trajectories and downstream hypotheses should be confirmed by multiple trajectory inference methods using different underlying assumptions."(15) Numerous algorithms are available for trajectory inference (e.g. Monocle, PAGA, Slingshot, RaceID/StemID, among many others) and their performance and suitability depends on the individual dataset and nature of the trajectories that are to be inferred. It is recommended to use dynGuidelines(16) for the selection of optimal pseudotime analysis methods. In the present manuscript, the authors do not provide any justification for their use of Monocle 3 over other trajectory inference approaches, nor do they employ a secondary trajectory inference method to confirm observations made with Monocle 3. Finally, follow-up validation experiments that the authors carry out have their own limitations and caveats (see below). Hence, while the microglial aging trajectories identified by this study are intriguing, they remain hypothetical trajectories that need to be proven with additional follow-up experiments.

We thank the reviewer for their suggestion. We have utilized the dynGuidelines kindly provided by the reviewer to utilize an additional trajectory inference tool to analyze our data. We selected Scorpius based on the structure of our data. The tool has provided additional support that microglia progress from a homeostatic state (*Cx3cr1*, *Mef2c*) to the induction of stress genes (*Hspa1, Atf3*) at an intermediate point during aging progression. Furthermore, we observe a concordant increase in ribosomal protein genes at a time point in the pseudotime analysis immediately prior to activation of inflammation-related genes (*Il1b, Cst7*). These additional analyses support the main findings of our original pseudotime analysis and have been added to the manuscript as Figure S3C,D. Additionally, in the statistical test that uncovers differentially expressed genes along the pseudotime trajectory in this analyses, we find that *Tgfb1* is one of the genes that is differentially expressed with peak expression at an intermediate timepoint along the pseudotime trajectory. Furthermore, we have done some preliminary trajectory analysis with slingshot (Street et al, BMC Genomics, PMID: 29914354) that found a similar trajectory with analogous gene expression patterns and dynamic expression of *Tgfb1*.

To follow up on the idea that TGFb1 signaling in microglia plays a key role in determining microglial aging trajectories, the authors use RNAscope to show that TGFb1 levels in microglia peak in middle age. They also treat primary LPS-activated microglia with TGFb1 and show that this restores expression of microglial homeostatic gene expression and dampens expression of stress response and, potentially, inflammatory genes. Finally, they utilize transgenic approaches to delete TGFb1 from microglia around 8-10mo of age and scRNAseq to show that homeostatic signatures are lost and inflammatory signatures are gained. Hence, findings in this study support the idea that TGFb1 can strongly regulate microglial phenotype. Loss of TGFb1 signaling to microglia in adulthood has already been shown to cause decreased microglial morphological complexity and upregulation of genes typically associated with microglial responses to CNS insults(17-19). TGFb1 signaling to microglia has also been implicated in microglial responses to disease and manipulations to increase this signaling can improve disease progression in some cases(19). In this light, the findings in the present study are largely confirmatory of previous findings in the literature. They also fall short of unequivocally demonstrating that TGFb1 signaling acts as a "checkpoint" for determining subsequent microglial aging trajectory. To show this clearly, one would need to perturb TGFb1 signaling around 12mo of age and carry out sequencing (bulkRNAseq or scRNAseq) of microglia at 18mo and 24mo. Such experiments could directly demonstrate whether the whole microglial population has been diverted to the TGFb1-low aging trajectory (that progresses through a translational burst state to an inflammation state as proposed). Future development of tools to tag TGFb1 high or low microglia could also enable fate tracing type experiments to directly show whether the TGFb1 state in middle age predicts cell state at later phases of aging.

We apologize for the use of the term “checkpoint” when referring to the role of *Tgfb1* in microglial aging. Instead, our model posits that Tgfb1 expression increases in response to the early insults of the aging process in an attempt to return microglia to homeostasis. Therefore, this would predict that increasing TGFB1 levels after an insult would decrease activation and age-related progression of microglia, which we demonstrate in vitro (Figure 3). Alternatively, the loss of TGFB1 should prevent microglia from returning to a homeostatic state after an age-related stressor, and thus increase the number of microglia in activated states. We observe this increase in activated microglia in our middle-aged microglia-specific Tgfb1 knockout mouse model. Furthermore, the haploinsufficiency of Tgfb1 at this age indicates that TGFB1 signaling in microglia is sensitive to relative levels of Tgfb1. The transient increase in *Tgfb1* expression further suggests that the threshold for TGFB1 signaling is dynamic. Finally, RNA-Seq analysis of both in vitro TGFB1 supplemented microglia and in vivo Tgfb1 depleted microglia highlight that TGFB1 alters the aging microglia transcriptome. Combined, these results provide evidence that *Tgfb1* modulates advancement of microglia through an aging continuum.

The present study would also like to draw links between features of microglial aging in the hippocampus and a decline in hippocampal-dependent cognition during aging. To this end, they carry out behavioral testing in 8-10mo old mice that have undergone microglial-specific TGFb1 deletion and find deficits in novel object recognition and contextual fear conditioning. While this provides compelling evidence that TGFb1 signaling in microglia can impact hippocampus-dependent cognition in midlife, it does not demonstrate that this signaling accelerates or modulates cognitive decline (see below). Age-associated cognitive decline refers to cognitive deficits that emerge as a result of the normative brain aging process (20-21). For a cognitive deficit to be considered age-associated cognitive decline, it must be shown that the cognitive operation under study was intact at some point earlier in the adult lifespan. This requires longitudinal study designs that determine whether a manipulation impacts the relationship between brain status and cognition as animals age (22-24). Alternatively, cross-sectional studies with adequate sample sizes can be used to sample the variability in cognitive outcomes at different points of the adult lifespan (22-24) and show that this is altered by a particular manipulation. For this specific study, one would ideally demonstrate that hippocampal-based learning/memory was intact at some point in the lifespan of mice with microglial TGFb1 KO but that this manipulation accelerated or exacerbated the emergence of deficits in hippocampal-dependent learning/memory during aging. In the absence of these types of data, the authors should tone down their claims that they have identified a cellular and molecular mechanism that contributes to cognitive decline.

We agree with the reviewer that to adequately demonstrate an age-dependent effect of microglia-derived TGFB1 on cognition it is necessary to perturb microglial TGFB1 at young and mature ages and assess the age-dependent effect on cognition. To address this, we have now performed a complementary behavioral study utilizing the Tmem119-CreER mouse model to drive the microglia-specific excision of *Tgfb1* in two separate cohorts of mice – one young (2-3 months) and one in mature mice (7-8 months) – followed by cognitive testing. Using the novel object recognition test, we find that young mice of all genotypes (WT, *Tgfb1* Het and *Tgfb1* cKO) retain the ability to recognize the novel object (as determined by having a significant preference in exploring the novel object). Alternatively, only the WT mature mice demonstrate a preference for the novel object, while the *Tgfb1* Het and *Tgfb1* cKO show no preference for the novel object. These behavioral data demonstrate an age-dependent necessity for microglia-specific TGFB1 in in maintain proper hippocampal-dependent memory and is now included in the manuscript as revised Figure 4I-J. We have also included additional behavioral tests (Y-Maze and open field) that did not show any difference between the genotypes as Figure S6D-G. Unfortunately, we were unable to perform the fear conditioning testing, as our apparatus broke during this time. Together, these results reveal that there is an age-dependent necessity for microglia-derived TGFB1 for hippocampal-dependent cognitive function.

A final point of clarification for the reader pertains to the mining of previously generated data sets within this study. The language in the results section, methods, and figure legends causes confusion about which experiments were actually carried out in this study versus previous studies. Some of the language makes it sound as though parabiosis experiments and experiments using mouse models of Alzheimer's Disease were carried out in this study. However, parabiosis and AD mouse model experiments were executed in previous studies (25,26), and in the present study, RNAseq datasets were accessed for targeted data mining. It is fantastic to see further mining of datasets that already exist in the field. However, descriptions in the results and methods sections need to make it crystal clear that this is what was done.

The reviewer makes an excellent point. While we referenced the public dataset in the original manuscript, the citation style of superscripted numbers diminishes our ability to adequately reference the datasets. Therefore, we have added the names of the first authors (Palovics for the parabiosis dataset and Sala Frigerio for the Alzheimer’s Disease dataset) to all the instances in the results and figure legends when we refer to these datasets.

Additional recommendations:Major comments.(1) There is some ambiguity surrounding how to interpret the microglial TGFb1 knockout that seems incompatible with viewing this molecule as a "checkpoint" in microglial aging. TGFb1 is believed to be primarily produced by microglia. Secreted TGFb1 is then detected by microglial TGFbR2. Are the microglia that have high levels of TGFb1 in middle age signaling to themselves (autocrine signaling)? Or contributing to a local milieu that impacts multiple neighbor microglia (paracrine signaling)? The authors could presumably look in their own dataset to evaluate microglial capacity to detect TGFb1 via its receptors.

We thank the reviewer for this insightful suggestion. We have undertaken analysis of our dataset to assess whether Tgfb1 acts through autocrine or paracrine signaling. To do so, we reanalyzed our microglia aging scRNA-Seq dataset leveraging the variation in microglia *Tgfb1* expression to probe the relative activity of TGFB1. Specifically, we partitioned microglia into quartiles based on their Tgfb1 expression, and subsequently investigated the expression of TGFB signaling effectors and targets. High expression of downstream TGFB signaling pathway components in microglia with high *Tgfb1* expression would point to autocrine mechanisms while, alternatively, high expression of downstream TGFB signaling pathway components in microglia with low *Tgfb1* expression would point to paracrine mechanisms. We observed highest expression of TGFB signaling pathway components and targets in microglia with the highest expression of *Tgfb1*. These data suggest that Tgfb1 acts through an autocrine mechanism. These results have been added to our manuscript as Figure S4E-G. Additionally, while our manuscript was under review, a paper by Bedolla et al (Nature Communications 2024; PMID: 38906887) was published that investigated the role of Tgfb1 in adult microglia. This paper utilized orthogonal techniques – sparse microglia-specific Tgfb1 knockout and IHC - to also suggest that microglia utilize autocrine Tgfb1 signaling. Together, these complementary data provide strong evidence that Tgfb1 acts through an autocrine mechanism in adult microglia.

(2) Conclusions of the study rest on the assumption that microglial inflammatory responses are a central driver of cognitive decline. They assume that manipulations that increase microglial progression into an inflammatory state will negatively impact cognitive function. Although there are certainly a lot of data in the field that inflammatory factors can impact synaptic function, additional experiments would be required to unequivocally demonstrate that a "TGFb1 dependent" progression of microglia to an inflammatory state underlies any observed changes in cognition. For example, in the context of microglial TGFb1 deletion, can NSAIDs or blockers of soluble TNFa (e.g. XENP345), or blockers of SPP1, etc. rescue behavior? Can microglial depletion in this context rescue behavior? Assuming behavior was carried out in the same microglial TGFb1 KO mice that were used for microglial scRNAseq, they could also carry out linear regression-type analyses to link microglial inflammatory status to the behavioral performance of individual mice. In the absence of additional evidence of this sort, the authors should tone down claims about mechanistic relationships between microglial state and cognitive performance.

We thank the reviewer for realizing that the link between cognition and inflammation in our paper is speculative. Therefore, we have taken the reviewer’s advice and toned down the claims linking inflammation to cognition in our manuscript. Instead, we connect the disruption in cognition to what is observed in our data, a loss of microglia homeostasis and a shift in the microglia aging trajectories.

Additional Recommendations:Minor comments:(1) Ideally at some point in the results or discussion, the authors should acknowledge that the hippocampus has highly distinct sub-regions and that microglia show different functions and properties across these sub-regions (e.g. microglia in hilus and subgranular zone vs microglia in stratum radiatum, vs microglia immediately adjacent to or embedded within stratum pyrimidale). Do expression levels of TGFb1 and microglial aging trajectories vary across sub-regions? To what extent can this account for heterogeneity of aging trajectories observed in microglial aging within the hippocampus?

We are interested in how microglia heterogeneity during aging is influenced by the specific functions, and thus microenvironments within the hippocampus. Therefore, we have expanded our IHC analysis of microglia to determine how the microenvironment influences microglia phenotypes by looking at several different regions of the hippocampus. We have included this regional analysis as Figure S2 in the manuscript. This analysis has revealed region-specific effects on microglia activation during aging.

(2) For immunohistochemistry data, it is not particularly convincing to see one example of one cell from each condition. Generally, an accepted approach in the field is to present lower magnification images accompanied by zoom panels for several cells from each field of view. This reassures the reader that specific cells haven't simply been "cherry-picked" to support a particular conclusion.

To allay the concerns of the reviewer that cells haven’t been “cherry-picked”, we have provided low magnification images for the aging CD68 and NF_κ_B stains in Supplemental Figure S2.

(3) In immunohistochemistry data, have measures been taken to ensure that observed signals are not simply autofluorescence that becomes prominent in tissues with aging? (i.e. use of trueblack or photoquenching of tissue prior to staining) See PMID 37923732

We agree that autofluorescence, at least partially due to the accumulation of lipofuscin, becomes prominent in certain regions and cells of the hippocampus during aging. This most prominently occurs in the microglia of the hilus. This autofluorescence has a particular subcellular distribution, as it is localized to lyso-endosomal bodies. The microglia activation marker CD68 is also localized to lysosomes. A previous publication by Burns et al (eLife; PMID: 32579115) identified autofluorescent microglia (AF+) with unique molecular profiles that accumulate with age. They posited that these AF+ microglia resembled other microglia subsets that have pronounced storage compartments, such as the pro-inflammatory lipid droplet-containing microglia that accumulate with age reported by Marschallinger et al (Nature; PMID: 31959936). As such, autofluorescence present in microglia potentially represents distinctive and functional states of microglia. Our CD68 immunostaining accumulates with age, which could overlap with autofluorescent storage bodies. Thus, we performed a complementary CD68 immunostaining in an independent cohort of young (3 months) and aged (24 months) mice with autofluorescence quencher TrueBlack, and found that the staining pattern and accumulation of CD68 microglia with age persisted as previously observed after use of this quencher (see Authpr response image 1). Images are IBA1 (cyan) and CD68 (yellow) with the molecular layer (ML), granule cell (GC), and hilus illustrated and corresponding quantification provided (Two-way ANOVA with Sidak’s multiple comparisons test; ***p<0.001; ****p<0.0001).

We would like to note that the subcellular localization of the other immunostainings included in the manuscript was distinct from CD68, and not likely to be associated with the autofluorescent storage bodies. Additionally, our RNAScope staining for *Tgfb1* did not show an accumulation with age, but rather a transient increase at 12 months of age, which indicates that the interpretation of the RNAScope stain for *Tgfb1* was not unduly influenced by autofluorescence.

**Author response image 1. sa2fig1:** 

(4) Ideally, more care is needed with the language used to describe microglial state during aging. The terms "dystrophic," "dysfunctional," and "inflammatory" all carry their own implications and assumptions. Many changes exhibited by microglia during aging can initially be adaptive or protective, particularly during middle age. Without additional experiments to show that specific microglial attributes during aging are actively detrimental to the tissue and additional experiments to show that microglia have ceased to be capable of engaging in many of their normal actions to support tissue homeostasis, the authors should exercise caution in using terms like dysfunctional.

We appreciate the reviewers’ suggestion. To allay the concerns of the reviewer about the multiple implications of terms such as “dysfunctional” and “inflammatory”, we have tried to replace them throughout the text with more specific terms.

**Reviewer #2:**
That said, given what we recently learned about microglia isolation for RNA-seq analysis, there is a danger that some of the observations are a result of not age, but cell stress from sample preparation (enzymatic digestion 10min at 37C; e.g. PMID: 35260865). Changes in cell state distribution along aging were made based on scRNA-seq and were not corroborated by any other method, such as imaging of cluster-specific marker expression in microglia at different ages. This analysis would allow confirming the scRNA-seq data and would also give us an idea of where the subsets are present within the hippocampus, and whether there is any interesting distribution of cell states (e.g. some are present closer to stem cells?). Since TGFb is thought to be crucial to microglia biology, it would be valuable to include more analysis of the mice with microglia-specific Tgfb deletion e.g. what was the efficiency of recombination in microglia? Did their numbers change after induction of Tgfb deletion in Cx3cr1-creERT2::Tgfb-flox mice.

We thank the reviewer for their comment regarding potential ex vivo transcriptional alterations with the approaches used in our study. We performed our aging microglia scRNA-Seq characterization prior to the release of Marsh et al (Nature Neuroscience; PMID: 35260865), which revealed the potential transcriptional artefacts induced by isolation. That being said, we took great care to minimize the amount of time samples were subjected to enzymatic digestion (15 minutes) and kept cells at 4C during the remainder of the isolation. Furthermore, we performed all isolations simultaneously, so that transcriptional changes induced by the isolation would be present across all ages and should not be observed during our analysis unless indicative of a true age-related change. Additionally, we have corroborated changes in cell state distribution across ages using several markers (*Tgfb1* and KLF2 for the intermediate stress state, S6 for the translation state, and NFKB and CD68 for activation states). In the revised manuscript, we have added additional hippocampal subregion analysis of several IHC immunostains to provide spatial insights into the microglia aging process (Figure S2). This analysis reveals unique spatial dynamics of microglia aging. For example, as the reviewer foresaw, we found that the granule cell layer (the location of adult hippocampal neurogenesis) had a more pronounced age-associated progression of microglial activation than several other regions. A subset of regions had minimal levels of activation during aging, such as the molecular layer and the stratum radiatum of the CA1 (inner CA1in the manuscript) – regions enriched in synaptic terminals. Furthermore, this analysis highlights the susceptibility of microglia aging to microenvironmental influences.

Regarding the temporally controlled microglia-specific genetic KO mouse model used in our original submission, the Cx3cr1-CreER allele selected (B6.129P2(Cg)-Cx3cr1tm2.1(cre/ERT2)Litt/WganJ) has been reported to have very high recombination efficiency (~94% in Parkhurst et al (Cell; PMID: 24360280)), and we used a tamoxifen induction protocol very similar to Faust et al. (Cell Reports; PMID: 37635351) that achieved ~98% recombination (they injected 100mg/kg for 5 days, while we injected 90mg/kg for 5 days). We analyzed our scRNA-Seq data for the expression of *Tgfb1* and found that the knockout mice had a 67% reduction in cells expressing higher levels of *Tgfb1* (see panel A in Author response image 2). This is likely a large underestimate of the recombination efficiency, as exon 3 is floxed and residual nonfunctional transcripts could be present, given nonsense-mediated decay is not realized in a number of knockout lines (Lindner et al, Methods, PMID: 33838271). We likely achieved a much higher excision efficiency. We would like to highlight that our data indicating increased microglia activation after tamoxifen treatment (Figure S5A) and the involvement of autonomous signaling (Figure S4E-G) are consistent with recently published work by Bedolla et al, (Nature Communications; PMID: 38906887). Additionally, as part of the revision process, we have now corroborated our behavioral data using and independent temporally controlled microglia-specific KO mouse model - Tmem119-CreER::Tgfb1 knockout mice (Figure 4I-K). We performed qPCR on sorted microglia to determine RNA levels in wildtype and knockout mice. Relative levels of *Tgfb1* and exon 3 of *Tgfb1* (the floxed exon) on technical replicates of 3 pooled samples indicated overall loss of *Tgfb1* expression, as well as undetectable levels of exon 3 as normalized to Actb (see panel B in Author response image 2).

With respect to the effects of aging and Tgfb1 on microglia density, we find a slight region-specific increase in microglia density with age (see Author response image 3). The density of Iba1 cells across hippocampal regions was analyzed at 3 and 24 months of age (see panel A in Author response image 3) and along an aging continuum at 3, 6, 12, 18, and 24 months (see panel B in Author response image 3). These data are also included in the revised manuscript (Figure S2D-F).

**Author response image 3. sa2fig3:** 

Deletion of Tgfb1 also had region-specific effects on microglia. While there was no difference in microglia density between wildtype and heterozygous microglia, there was a significant increase in microglia density in the hilus and molecular layers in knockout mice (see Author response image 4) and included in the revised manuscript (Figure S5A). These data indicate that there are subtle region-specific increases in microglia density with age, as well as following the deletion of *Tgfb1* from microglia of mature mice.

**Author response image 4. sa2fig4:** 

Additional Recommendations:(1) The problem of possible digestion artifacts in scRNA-seq should be at least addressed in the discussion as a caveat in data interpretation. Staining for unique cluster markers in undigested tissue would solve the problem. It can be done with microscopy or using flow cytometry, but for this microglia, isolation should be done with no enzymes or with Actinomycin (PMID: 35260865).

The ex vivo activation signature uncovered by Marsh et al. (Nature Neuroscience; PMID: 35260865) arises from the digestion methods used to isolate microglia. We took the utmost care in processing our microglia identically within experiments, which should minimize the amount of uneven ex vivo activation of microglia. This is borne out by the structures of our single-cell sequencing data. Unlike Marsh et al_._ where they observe unique cluster after addition of their inhibitors, we do not see any clusters unique to a single condition, suggesting that any influence of ex vivo activation was evenly distributed.

Importantly, as suggested by the review, we have we have complemented our scRNA-Seq analysis by corroborating several markers for various stages of microglia aging progression using RNAScope and IHC in intact tissue. Specifically, the transient age-dependent increase in Tgfb1 high microglia was confirmed using RNAScope (Figure 3B), the age-related increase in ribosomal high microglia was confirmed using S6 immunostaining (Figure 3I), and the increase of various markers of age-associated activation (C1q, CD68 and NFkB) was confirmed using immunostaining (Figure 1F and Figure S2D-I). Additionally, we have also performed immunostainings for KLF2 and confirmed peak microglia expression at 18 months of age with lower levels at 24 months of age (Figure 2H).

(2) The figures of GO and violin plots are not easy to follow sometimes... what are the data points in the violin plots, maybe worth showing them as points? For the GO, e.g. in 3D, 3J, including a short description of the figure could help, e.g. in Figure 1. it was clear.

We chose not to include the datapoints in the violin plots for aesthetic purposes. Each violin plot would have had hundreds of points that would have made the plots very busy and hidden the structure of the distribution. In Author response image 5 we show the violin plot in Figure 2M with (panel A) and without (panel B) individual points. In a small format, the points overlap and become jumbled together. Therefore, we chose to present the violin plots without points for clarity on the data structure. As for the gene ontology plots in Figure 3, we have updated the descriptions in both the text and figure legends to provide clarification on what they represent.

**Author response image 5. sa2fig5:** 

(3) I'm very curious to see the mechanism of action of "aged" microglia in the TGFb-depletion model. Is it creating hostile conditions for stem cells, or we have increased synapse loss? Something else?

We thank the reviewer for their insightful questions. We would like to note that during the revision process of our manuscript, a complementary study was published reporting that the loss of microglia-derived Tgfb1 leads to an aberrant increase in the density of dendritic spines in the CA1 region of the hippocampus (Bedolla et al, Nature Communications, PMID: 38906887). The data from Bedolla et al, shows sparsely labeled neurons in the CA1 with a mGreenLantern expressing virus in mice the had Tgfb1 deleted from microglia using the Cx3cr1-CreERT driver (Figure 7U,V). Additionally, McNamara et al (Nature; PMID: 36517604) demonstrated that microglia-derived Tgfb1 signaling regulates myelin integrity during development and several studies have revealed links between Tgfb1 signaling and altered neurogenesis (e.g., He et al, Nature, PMID: 24859199 and Dias et al, Neuron, PMID: 25467979). Together, this growing body of work indicates that microglia-derived TGFB1 regulates myelination, neurogenesis and synaptic plasticity, which have all been shown to play a role in cognition.